# Copy number variation of the restorer *Rf4* underlies human selection of three-line hybrid rice breeding

Zhe Zhao[1,2,4], Zhi Ding[1,2,4], Jingjing Huang[1,2], Hengjun Meng[1,2], Zixu Zhang[1,2], Xin Gou[1,2], Huiwu Tang[1], Xianrong Xie[1,2], Jingyao Ping[1], Fangming Xiao[3], Yao-Guang Liu [1,2], Yongyao Xie[1,2] ✉ & Letian Chen [1,2] ✉

Cytoplasmic male sterility (CMS) lines are important for breeding hybrid crops, and utilization of CMS lines requires strong fertility restorer (*Rf*) genes. *Rf4*, a major *Rf* for Wild-Abortive CMS (CMS-WA), has been cloned in rice. However, the *Rf4* evolution and formation of CMS-WA/*Rf* system remain elusive. Here, we show that the *Rf4* locus emerges earlier than the CMS-WA gene *WA352* in wild rice, and 69 haplotypes of the *Rf4* locus are generated in the *Oryza* genus through the copy number and sequence variations. Eight of these haplotypes of the *Rf4* locus are enriched in modern rice cultivars during natural and human selections, whereas non-functional *rf4i* is preferentially selected for breeding current CMS-WA lines. We further verify that varieties carrying two-copy *Rf4* haplotype have stronger fertility restoration ability and are widely used in three-line hybrid rice breeding. Our findings increase our understanding of CMS/*Rf* systems and will likely benefit crop breeding.

Cytoplasmic male sterility (CMS), a maternally inherited trait caused by mitochondrial gene rearrangement and the interaction between mitochondrial genes and nuclear genes, occurs widely in plants[1–3]. The discovery of CMS and fertility restorer (*Rf*) genes in rice (*Oryza sativa*) enables the production of hybrid rice using a three-line system, a successful model for utilizing heterosis to improve rice yield[3]. Three major types of CMS systems have been used in three-line hybrid rice breeding: Wild Abortive type (CMS-WA)[4], HongLian type (CMS-HL)[5], and Boro II type (CMS-BT)[6]. Three-line hybrid rice production in China primarily uses CMS-WA rather than CMS-HL and CMS-BT (1983–2018)[7,8] (http://www.ricedata.cn/). Because three-line hybrid rice production requires restoration of fertility in CMS, understanding the evolution and selection of related *Rf* genes provides important information for facilitating and potentially improving hybrid rice breeding.

The *Oryza* genus, including 25 wild species and 2 cultivated species (Asian cultivated rice *O. sativa* and African cultivated rice *O. glaberrima*), has undergone significant diversification over the course of evolution, with many of *Oryza* species diverging within the past 15 million years[9,10]. Asian cultivated rice is divided into two subspecies: *O. sativa indica* and *O. sativa japonica*[11–13]. The CMS-WA lines were developed by introgressing CMS cytoplasm from common wild rice (*O. rufipogon*) into some *indica* rice cultivars[4].

We previously cloned the CMS-WA gene *WA352*, which encodes a cytotoxic mitochondrial protein that causes premature programmed cell death in anther tapetum cells, ultimately leading to pollen abortion[14]. According to our evolutionary analysis, the mitochondrial gene *WA352* (renamed *WA352c*[1]) is a newly formed CMS gene derived from non-functional CMS-like protogenes (*orf356* and *orf367*), through multiple rearrangements of mitochondrial genome sequences accompanied by sub-stoichiometric shifting, sequence variation, and natural selection in *O. rufipogon*[1]. We also cloned one of the major CMS-WA restorer genes, *Rf4*[15]; the other CMS-WA restorer gene, *Rf3*, remains to be cloned[16]. *Rf4* encodes a pentatricopeptide repeat (PPR) protein of 782 amino acids, which restores CMS-WA fertility by mediating degradation of the *WA352* mRNA[15]. Although we and other

[1]State Key Laboratory for Conservation and Utilization of Subtropical Agro-Bioresources, College of Life Sciences, South China Agricultural University, Guangzhou 510642, China. [2]Guangdong Laboratory for Lingnan Modern Agriculture, Guangzhou 510642, China. [3]Department of Plant Sciences, University of Idaho, Moscow, ID 83844, USA. [4]These authors contributed equally: Zhe Zhao, Zhi Ding. ✉e-mail: yyxie@scau.edu.cn; lotichen@scau.edu.cn

groups previously identified five variants of *Rf4* (*Rf4^M* [haplotype H1], *Rf4^I* [H2], H3 [*rf4aus*, see below], *rf4i* [H4], and *rf4j* [H5])[15,17,18], the evolutionary trajectory of *Rf4* in the *Oryza* genus is unclear, and how the CMS-WA gene *WA352c* co-evolved with the functional *Rf4* gene during natural and human selections remains unknown.

Most *Rf* genes in plants encode P-type PPR proteins function as sequence-specific RNA binding proteins to destroy the functions of newly emerging mitochondrial CMS genes, a process that might be driven by selection[19–21]. *PPR* genes often occur in clusters and *PPR* genes within these clusters show high sequence similarity; these clusters may form through tandem duplication, thus contributing to expansion of the *PPR* gene family[22]. For example, a *PPR* gene cluster on rice chromosome 10 consists of at least ten *PPR* genes within an about 400-kb region (according to the reference genome of a restorer line MH63 (Minghui63, fertility restorer line carrying functional *Rf4*, http://rice.hzau.edu.cn/rice_rs3/)[23], including *Rf4* for CMS-WA[15], *Rf1a* and *Rf1b* for CMS-BT[24], *Rf5* for CMS-HL[25], and *Rf19* for CMS-FA[26]. The *PPR* genes in the *Rf4* locus have evolved rapidly through sequence duplication and allelic variation, generating functional *Rf4* variants and non-functional *rf4* variants[15]. Nonetheless, whether the *Rf4* locus also underwent copy number variation (CNV) remains unclear.

CNV is a significant source of genetic variation, shapes gene expression profiles, and contributes to various adaptive traits in plants[27,28]. CNV of genes usually affects gene expression via a gene dosage effect (i.e. more copies of the gene produce more transcripts, which then produce more protein), thus affecting phenotype[29,30]. For example, CNV in the *GL7* locus in rice affects grain size[31]. CNV of *TdDof* controls hollow- and solid-stemmed structure in different wheat (*Triticum* spp.) species[32]. CNV at the *WUS1* locus and the *Fas1* locus controls the fate of inflorescence stem cells in maize (*Zea mays*)[33,34]. In sugar beet (*Beta vulgaris*), gene dosage of the semidominant locus *Rf1* (heterozygotes and homozygotes) affect fertility restoration[35]. Although different CNVs have dosage effects on different agronomic traits in crops[36,37], to date, whether CNV-mediated gene dosage of *Rf* genes affects fertility restoration remains unclear.

In this work, we investigate the origin and evolution of *Rf4* in numerous wild rice accessions, landraces, and cultivars. We identify diverse structural variations (SV) and CNV at the *Rf4* locus. We find that the CNV of functional *Rf4* contributes to fertility restoration of CMS-WA in a dosage-dependent manner. We also reveal the evolutionary history of the *Rf4* locus and the co-evolution of *Rf4* haplotypes and the CMS gene *WA352c*. In addition, we develop a set of molecular markers for screening desirable *Rf4* haplotypes for breeding. Our findings refine our understanding of the evolutionary plasticity of the *Rf* genes and the formation of the CMS/*Rf* systems, which may accelerate the breeding of elite rice varieties as strong restorer lines for hybrid rice production.

## Results
### Identification of variants and haplotypes of the *Rf4* locus in rice cultivars
Our and others previous studies identified five variants at the *Rf4* locus, namely, *Rf4^M* and *Rf4^I* (in *indica* fertile restorer lines), *rf4aus* (in circum-*aus* rice, see below), *rf4i* (in *indica* CMS-WA/maintainer lines), and *rf4j* (in *japonica* rice)[15,17,18]. To further reveal the origin and evolution of the *Rf4* locus, we re-analyzed the genome sequences of the *Rf4* locus using publicly available rice genome information for ZS97B (Zhenshan97B, a maintainer line with *rf4i*) (http://rice.hzau.edu.cn/rice_rs3/)[23], Nip (Nipponbare, a *japonica* line with *rf4j*) (https://rapdb.dna.affrc.go.jp/)[38,39], and MH63[23] and SH498 (https://www.mbkbase.org/rice)[40] (Shuhui498, fertility restorer line carrying functional *Rf4*). We focused on part of a *PPR* cluster that includes the *Rf4* locus based on the reference genome of the restorer line MH63 (Fig. 1a).

At the *Rf4* locus region, we first examined two lines that lack the ability to restore fertility and identified three *PPR* genes (*PPR7*

[*Os10g0495400*], *rf4j* [*Os10g0495200*], and *PPR10* [*Os10g0495100*]) in Nip (Fig. 1a) and three *PPR* genes (*PPR7*, *rf4i*, *PPR10*) in ZS97B (Fig. 1a). The *rf4i* variant was pseudogenized and non-functional due to the presence of a premature termination codon (Fig. 1a).

We next examined two lines that have the ability to restore fertility. The *Rf4* locus regions of MH63 and SH498 possess seven *PPR* genes, in addition to previously known functional *Rf4* variant (here defined as *Rf4a*, of functional *Rf4* in Copy-a site of *Rf4* locus), *PPR7*, *PPR8*, and three copies of *PPR10* genes, which included another functional *Rf4* variant *Rf4b* identified in the Copy-b site of *Rf4* locus (Fig. 1a). The interval between *Rf4a* and *Rf4b* is 74.8 kb, and the *Rf4a* and *Rf4b* coding sequences show 100% amino acid identity. The regions 7.5 kb upstream of the start codon and 1.5 kb downstream of the stop codon of *Rf4a* and *Rf4b* showed 98.4% and 99.7% similarity, respectively (Supplementary Data 1). These findings reveal that different rice varieties show SV and CNV at the *Rf4* locus.

Then, we further investigated SV and CNV at the *Rf4* locus region in 311 rice cultivars by PCR amplification and sequencing. To this end, we designed site-specific PCR primers based on single-nucleotide polymorphisms (SNPs, Supplementary Table 1). The *Rf4a* and *Rf4b* genes were amplified using a common primer F1 combined with the site-specific reverse primers a-R and b-R, respectively; the *rf4i* fragment was amplified using a primer pair F2 and i-R (Fig. 1a). In total, there were seven variants and eight haplotypes (H1–H8) based on the combination of the Copy-a and Copy-b variants at the *Rf4* locus in the modern rice cultivars (Fig. 1b and Supplementary Data 2). Sequence analyses demonstrated that the restorer lines contain five haplotypes with one- or two-copy *Rf4* variants, including *Rf4a^I* (H2), *Rf4b^M* (H6), *Rf4a^M-Rf4b^M* (H1), *Rf4a^I-rf4b* (H7), and *rf4a-Rf4b^M* (H8), while all current CMS-WA lines (and their maintainer lines) carry the *rf4i* (H4) variant; the *rf4j* (H5) variant is present mainly in *japonica* cultivars. Among these variants, *rf4aus* was previously named as H3[18] (Fig. 1b).

### Diversity and distribution of *Rf4* and *rf4* variants and haplotypes in the *Oryza* genus
To trace the evolutionary history of *Rf4* in the *Oryza* genus, we further investigated the CNV and sequence polymorphisms of *Rf4* and *rf4* variants among wild rice and landrace rice accessions. In addition to the seven *Rf4* and *rf4* variants identified in the cultivars, 61 variants were identified in the Copy-a and/or Copy-b sites of the *Rf4* locus (from GG to AA-genome species) (Supplementary Data 2–5), pointing to the prevalence of SV and CNV in the *Rf4* locus.

We then performed a BLAST search for putative orthologs and homologs of *Rf4* and *rf4* in Poaceae genomes in the GenBank database (https://www.ncbi.nlm.nih.gov/). The putative homologs in *Aegilops tauschii*, *Setaria italica*, *Setaria viridis*, *Sorghum bicolor*, *Triticum aestivum*, *Triticum dicoccoides*, and *Zea mays* were similar to the rice *Rf4* and *rf4* variants (Supplementary Fig. 1). The homologous gene *LOC117839145* of *S. viridis* shared the highest nucleotide sequence similarity of 72.2% with the *Rf4^M* variants in the rice cultivars (Supplementary Data 6).

We next used the nucleotide sequences to construct a phylogenetic tree of *Rf4* and *rf4* in the *Oryza* genus using *LOC117839145* (*S. viridis*) as the outgroup. The variants of different species were divided into six homologous lineages (Supplementary Fig. 2). Among these, the *rf4i*, *rf4a*, and *rf4b* clades were closely related to each other, and *Rf4^M* and *Rf4^I* were in the same clade, whereas the lineages containing *rf4aus* and *rf4j* were closely related (Supplementary Fig. 2). During the evolution of wild rice, a considerable number of *rf4a*, *rf4b*, and *Rf4* variants were developed, but only two variants each of *rf4j* and *rf4i* were generated (Supplementary Fig. 2). Among *Rf4* variants in the different rice cultivars, *rf4a*, *rf4b*, *rf4aus*, *rf4i*, and *Rf4* mainly occur in the *indica* group, *rf4j* mainly occurs in the *japonica* group, and *rf4i* was only found in bred maintainer lines and CMS-WA lines (Supplementary Data 2).

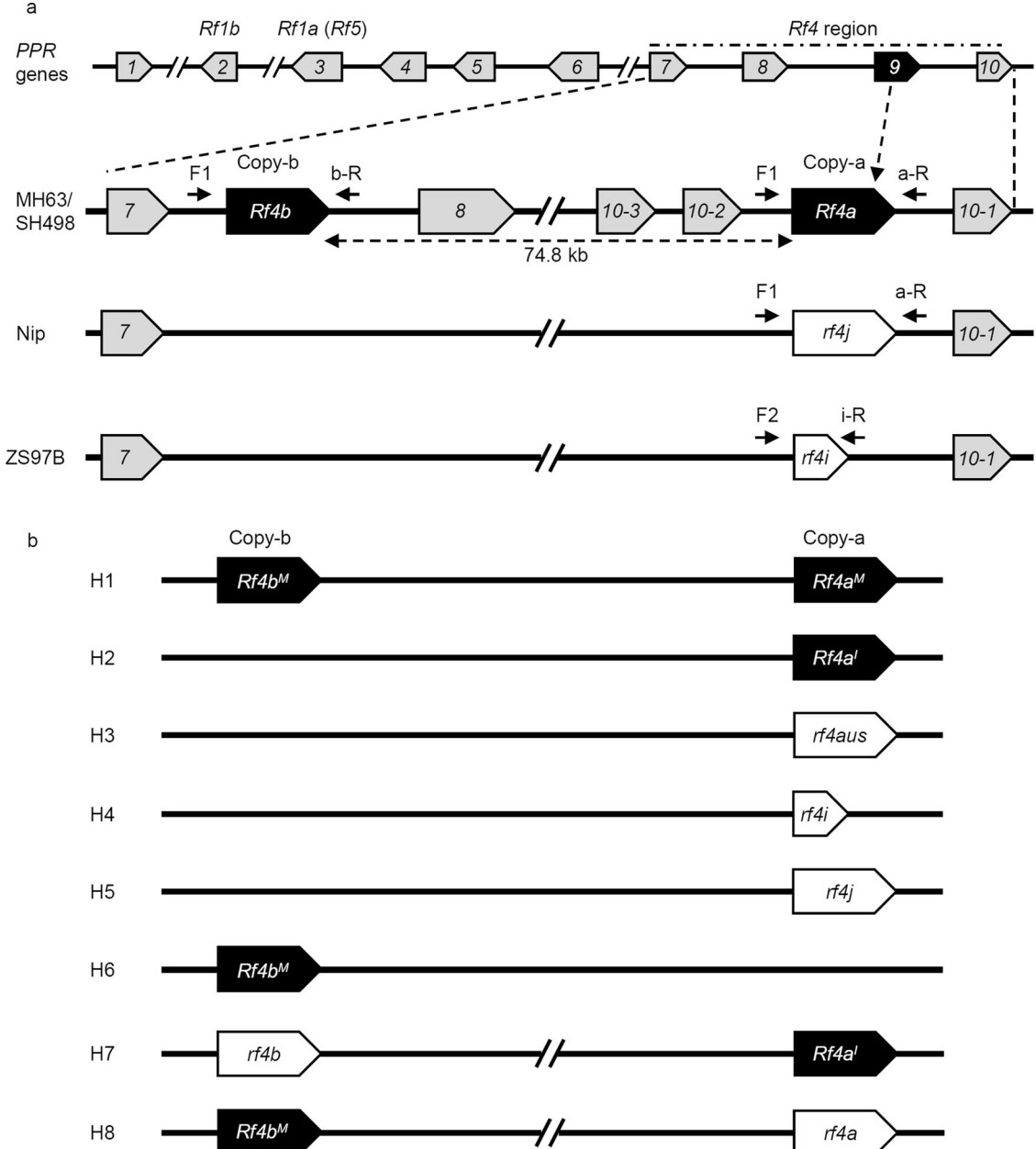

**Fig. 1 | Genomic structure variations and haplotypes at the *Rf4* locus.**
**a** Homologous gene relationship (microsynteny block) at the *Rf4* locus in *O. sativa* ssp. *indica* and *O. sativa* ssp. *japonica*. **b** Different haplotypes of the *Rf4* complex locus region involving in the *Rf4a*-copy (Copy-a) and *Rf4b*-copy (Copy-b) identified in Asian cultivated rice. Gray background shows other *PPR* genes, white background shows non-functional *rf4* variants, black background shows functional *Rf4* variants. *j*: *japonica*, *aus*: circum-*aus*, *i*: *indica*, I: IR24, M: MH63, Nip: Nipponbare (a *japonica* variety). IR24 and MH63 are *indica* restorer lines.

In the wild rice *O. meyeriana* (GG-genome), only a putative one-copy *Rf4* variant appeared at the Copy-a site, but not the Copy-b site, thus, this Copy-a variant sequence likely represents the primitive sequence of ancestral *Rf4*, which we named *Anc-Rf4* (Fig. 2 and Supplementary Data 3). Along with sequence variation and genetic recombination, various haplotypes (H13-H68) consisting of *Rf4-like* and/ or *rf4-like* variants were generated in the genomes of EE (*O. australiensis*), CCDD (*O. alta*, *O. grandiglumis*, and *O. latifolia*), CC (*O. eichingeri*, *O. officinalis*, and *O. rhizomatis*), BBCC (*O. minuta*), BB (*O. punctata*), and AA genomes (*O. meridionalis*, *O. glumaepatula*, *O. rufipogon*, and *O. nivara*) of these wild species (Fig. 2 and Supplementary Data 3); including one-copy haplotypes *Rf4a/rf4a-likes* (a group of *Rf4a-like* or *rf4a-like* variants with SNPs) at the Copy-a site, and *Rf4b/rf4b-likes* (a group of *Rf4b-like* or *rf4b-like* variants with SNPs) at the Copy-b site (H13-H27) (Fig. 2 and Supplementary Data 3). Further, various two-copy

haplotypes containing *Rf4-like* and/or *rf4-like* variants were formed, including *Rf4a^M-rf4b-likes*, *rf4a-likes-Rf4b^M*, *rf4a-likes-Rf4b-likes*, *Rf4a-likes-Rf4b-likes*, and *rf4a-likes-rf4b-likes* (H28-H68) (Fig. 2 and Supplementary Data 3). However, no *Rf4* and *rf4* variants were detected in the tested *O. longistaminata*, *O. barthii*, and *O. glaberrima* accessions at the Copy-a or Copy-b sites (Fig. 2 and Supplementary Data 2 and 3). The variants such as *rf4i*, *Rf4a^I*, and *rf4b* were first identified in *O. australiensis* (EE-genome), while *Rf4^M* and *rf4j* were first identified in *O. officinalis* (CC-genome), and *rf4aus* and *rf4a* appeared only in *indica* in *O. sativa* (Fig. 2 and Supplementary Data 2–4).

**The *rf4* haplotypes are non-functional in restoration of fertility in CMS-WA**
To confirm the biological functions of the *rf4* haplotypes in the rice cultivars, we analyzed amino acid sequence variation among six

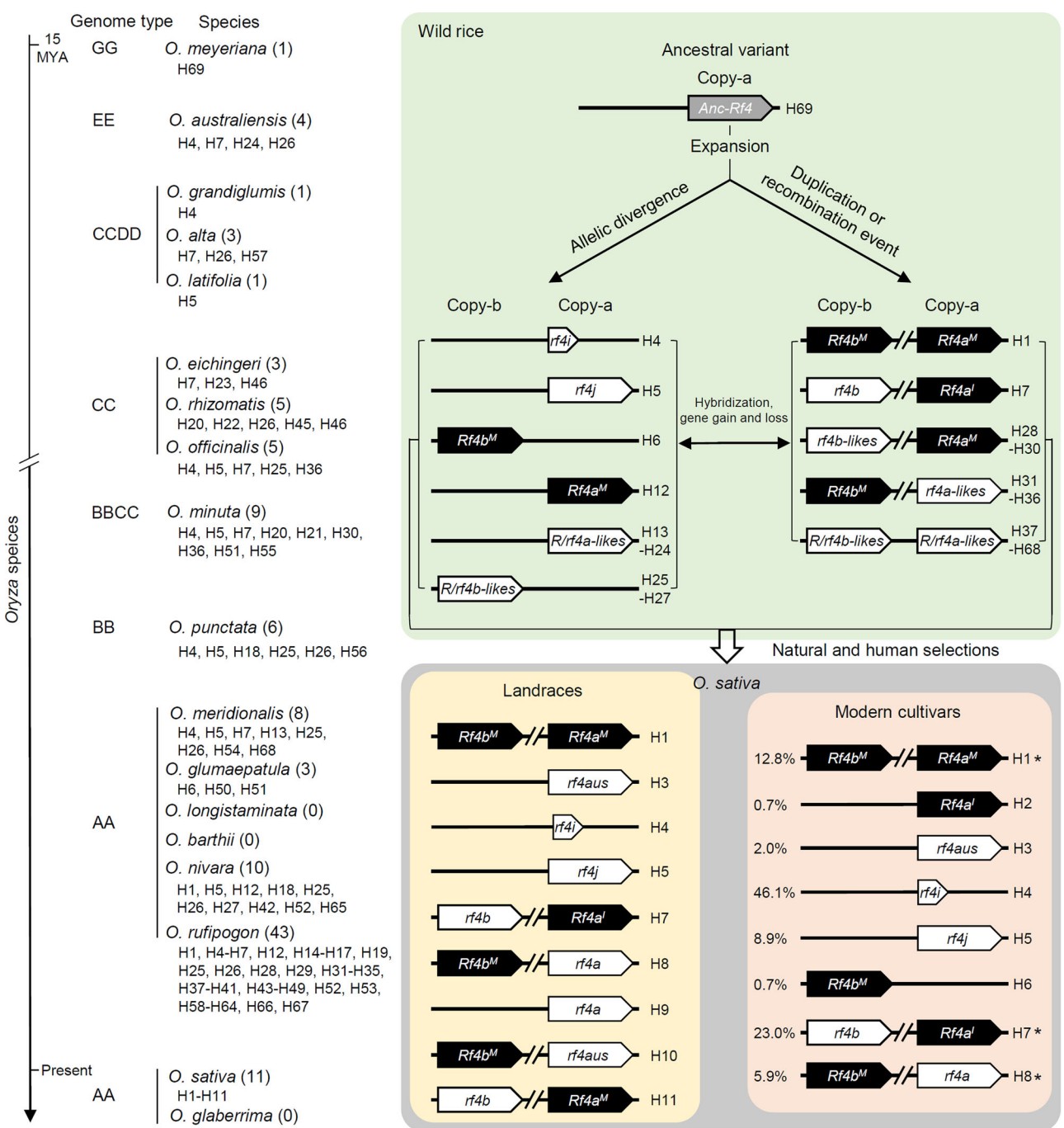

**Fig. 2 | Evolutionary trajectory of *Rf4* haplotypes in *Oryza* species.** Current *Rf4* haplotypes may have originated from an ancestral type of *Rf4* (*Anc-Rf4*, H69), which first emerged at Copy-a in the oldest wild rice *O. meyeriana* (GG-genome). Sequence variation, gene duplication, and recombination events resulted in new one-copy (H12-H27) or two-copy (H28-H68) haplotypes of *R/rf4a-likes* (a group of *Rf4a-like* or *rf4a-like* variants with SNPs) and/or *R/rf4b*-likes (a group of *Rf4b-like* or *rf4b-like* variants with SNPs) in wild rice. During evolution, along with natural and human selections, the nascent one-copy and two-copy *Rf4* and/or *rf4* haplotypes gradually migrated into the lineages of *O. rufipogon*, *O. nivara*, and *O. sativa*. *Rf4* haplotypes were not detected in tested accessions of *O. longistaminata*, *O. barthii*, and *O. glaberrima*. Among the eight *Rf4* haplotypes (H1–H8) in modern cultivars,

three two-copy haplotypes (indicated by asterisks "*", the percentages on the left showed frequency of specific haplotype in the tested accessions) are predominant in restorer lines. H1 (*Rf4a^M^-Rf4b^M^*) and H7 (*Rf4a^I^-rf4b*) haplotypes first appeared in *O. nivara* and *O. australiensis* (EE-genome), respectively, while H8 (*rf4a-Rf4b^M^*) is present only in landraces and modern cultivars of *O. sativa*. The one-copy variant *Rf4a^I^* in H2 of modern cultivars was only found in H7, suggesting that H2 may be derived from loss of *rf4b* in H7 from *O. rufipogon*. Other one-copy haplotypes such as H3 (*rf4aus*), H4 (*rf4i*), H5 (*rf4j*) and H6 (*Rf4b^M^*) first emerged in *O. sativa*, *O. australiensis*, *O. officinalis* (CC-genome), and *O. glumaepatula* (AA-genome), respectively. The number in the brackets next to each species represents frequency of the haplotypes detected in the species. MYA: million years ago (divergence time).

proteins: rf4a, rf4b, rf4j, rf4aus, Rf4^M^, and rf4i. Whereas rf4i is a truncated product containing only eight PPR motifs, the other five proteins contain 18 PPR motifs and share high sequence similarity (93.5–95.3%) (Supplementary Data 7). The rf4j, rf4b, rf4aus, and Rf4^M^ proteins contain 782 amino acids, while rf4a has 798 amino acids

(Supplementary Fig. 3). In contrast to rf4j *vs.* Rf4^M^, which harbor 37 amino acids differences[15,18], we detected 68 amino acid differences between rf4a and Rf4^M^, including two amino acid insertions at the N-terminal regions and 14 amino acid insertions at the C-terminal regions. In addition, rf4b has 38 amino acid differences, and rf4aus

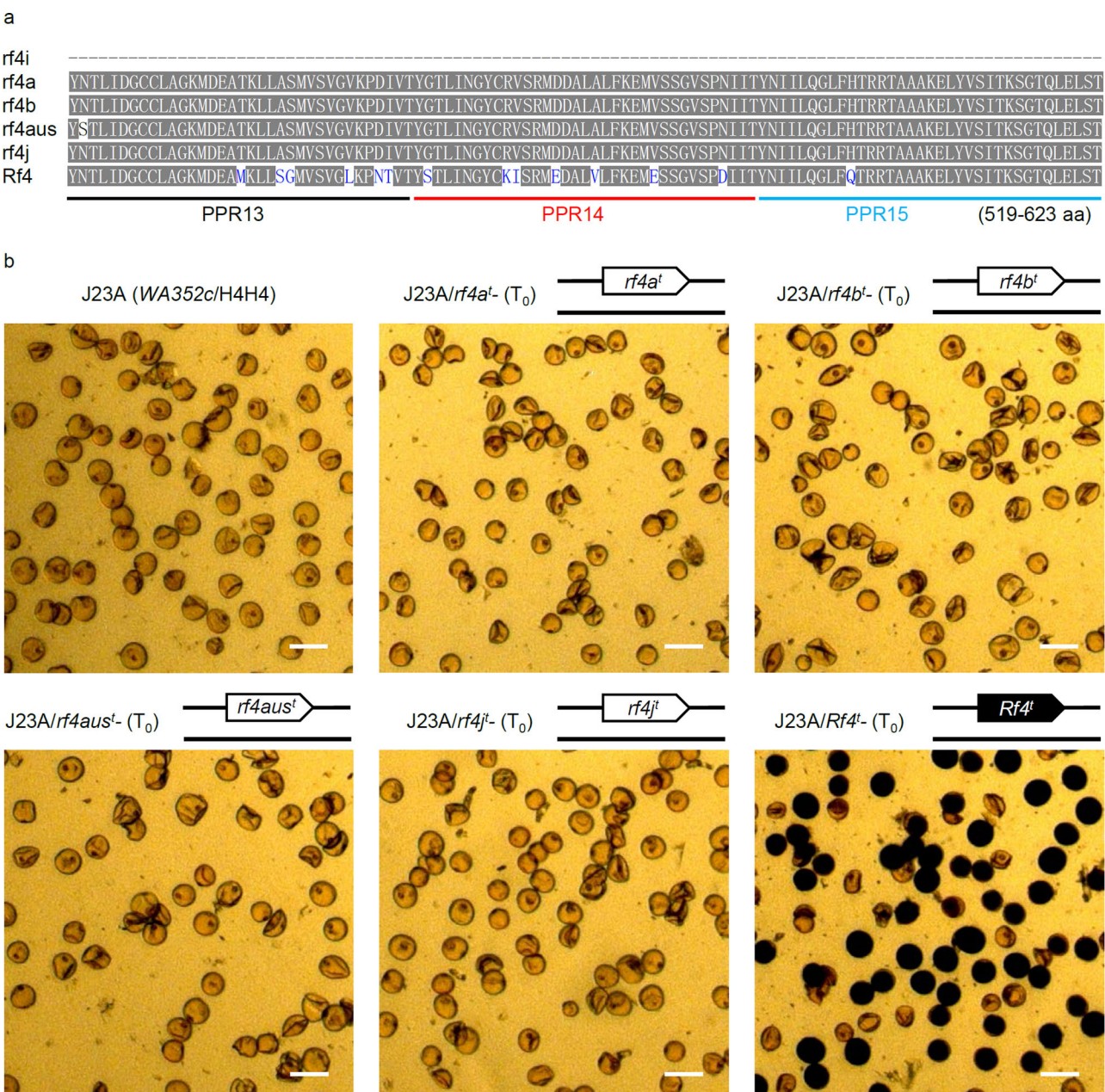

**Fig. 3 | Functional characterization of *Rf4* haplotypes in cultivated rice. a** Amino acid differences used to identify the functional Rf4 and non-functional rf4 variants in three PPR motifs (PPR13, PPR14, and PPR15). **b** Male fertility of transgenic (T₀) lines of Jin23A (a CMS-WA line) carrying different transgenes (ᵗ) in the hemizygous state. For each complementary construct, at least 10 independent transgenic lines with similar phenotype were obtained. The pollen phenotype of three independent lines was shown in Fig. 3b and Supplementary Fig. 4. Viable pollen stain black; inviable pollen stain light brown. Scale bar: 50 μm.

harbors 51 amino acid differences to Rf4$^M$ (Supplementary Fig. 3). Compared to Rf4$^M$, all rf4 proteins contain the common 14 amino acid substitutions at the PPR13, PPR14, and PPR15 motifs (Fig. 3a and Supplementary Fig. 3), suggesting that amino acid substitutions at these PPRs motifs are important for the fertility restoration of CMS-WA.

To verify our hypothesis that these *rf4* variants may be non-functional for fertility restoration, binary vectors containing *rf4a*, *rf4b*, *rf4j*, *rf4aus* and *Rf4$^M$*, all driven by the native promoter from *Rf4*, were constructed and introduced into the CMS-WA line Jin23A (*WA352c*/H4H4) via transformation. The male fertility of transgenic T₀ plants with the transgene containing *Rf4$^M$* was partially restored (Fig. 3b and Supplementary Fig. 4). Nevertheless, T₀ transgenic plants carrying *rf4a*, *rf4b*, *rf4j*, or *rf4aus* remained completely sterile (Fig. 3b and Supplementary Fig. 4). These results confirm the notion that all the *rf4* haplotypes are non-functional for fertility restoration of CMS-WA.

## CNV of *Rf4* has a dosage effect on fertility restoration of CMS-WA

We demonstrated that the *Rf4* locus has undergone CNV in different rice cultivars, and assumed that CNV at the *Rf4* locus may be associated with the effect on fertility restoration of CMS-WA. To verify the CNV-mediated gene dosage effect of *Rf4*, we produced various lines with different copy numbers of *Rf4* by crossing Jin23A (*WA352c*/H4H4) with two near-isogenic lines of *Rf4*: ZSRf4I (*WA352c*/H7H7) and ZSRf4M (*WA352c*/H1H1), knocking out *Rf4* in ZSRf4M lines, and transforming Jin23A with functional *Rf4*, respectively (Figs. 4, 5 and Supplementary Figs. 5, 6). Firstly, the pollen viability (assessed by staining with I₂–KI) of the F₁ plants derived from Jin23A×ZSRf4I (*WA352c*/H4H7) and Jin23A×ZSRf4M (*WA352c*/H4H1) were ~71% and ~88%, respectively, while the seed setting rates of these F₁ plants were ~34% and ~52%, respectively (Fig. 4a and Supplementary Figs. 5a, 7a, b), showing the

male fertility of the two-copy *Rf4*-carrying plants was higher than those of the one-copy *Rf4*-carrying plants (Fig. 4 and Supplementary Figs. 5, 7). Moreover, the fertile anthers were pollen-filled and yellowish, while the sterile anthers appeared thin and whitish (Figs. 4, 5 and Supplementary Figs. 5, 6).

Furthermore, we knocked out both *Rf4a^M* and *Rf4b^M* in ZSRf4M line by CRISPR/Cas9 editing and obtained several *rf4a^m* and *rf4b^m* mutants, which showed full abortion of pollen and spikelet (Fig. 4b, Supplementary Fig. 5b, and Supplementary Table 2). Then, we selected three *rf4a^m* and *rf4b^m* loss-of-function mutant lines (carrying different editing patterns) and crossed them with wild type ZSRf4M to test the dosage effect of *Rf4* in the resultant mutant F$_1$ (mF$_1$) (Fig. 4b and Supplementary Fig. 5b). As expected, the pollen and spikelets of mF$_1$ plants carrying two copies of *Rf4* (*WA352c*/H1h1) also showed lower pollen viability (~85%) and seed setting rate (~48%), compared to those of wild type ZSRf4M (~92% and ~72%, respectively), which carries four copies of *Rf4* (Fig. 4b and Supplementary Figs. 5, 7c, d).

To clarify the connection between the *Rf4* CNV-mediated gene dosage effect and the *WA352c* repression in fertility restoration, we performed qRT-PCR analysis of *Rf4* and *WA352c* expression in anthers of different plants at the microspore mother cell stage. The expression level of *Rf4* was twice as high in Jin23A×ZSRf4M (*WA352c*/ H4H1) *vs.* Jin23A×ZSRf4I (*WA352c*/H4H7) (Fig. 4c and Supplementary Fig. 5c), whereas the expression pattern of *WA352c* was opposite to that of *Rf4* (Fig. 4d and Supplementary Fig. 5d). The level of *WA352c* transcripts was higher in the *rf4a^m*-*rf4b^m*-mF$_1$ and *rf4a^m*-*rf4b^m* lines compared to ZSRf4M (Fig. 4e and Supplementary Fig. 5e).

In addition, we generated *Rf4^t*-transgenic lines using CMS line Jin23A as recipient and selected a homozygous Jin23A/*Rf4^tRf4^t* (*WA352c*/ H4H4*Rf4^tRf4^t*) plant from the T$_1$ population to cross with ZSRf4I generating F$_1$ plants Jin23A/*Rf4^tRf4^t*×ZSRf4I (*WA352c*/H4H7*Rf4^t-*). We then acquired a series of materials harboring different copy numbers of the *Rf4^t* transgene in the F$_2$ population. As expected, lines with two-copy *Rf4*, including Jin23A/*Rf4^tRf4^t* (*WA352c*/H4H4*Rf4^tRf4^t*) and Jin23A/ *Rf4^tRf4^t*×ZSRf4I (*WA352c*/H4H7*Rf4^t-*), exhibited higher pollen viability (~87%, ~90%) and spikelet fertility (~52%, ~53%) than J23A/*Rf4^t-* lines harboring a single copy of *Rf4* (*WA352c*/H4H4*Rf4^t-*), which showed ~73% pollen fertility and ~36% spikelet fertility (Fig. 5a and Supplementary Figs. 6a, 7e, f).

The level of *Rf4* transcripts in Jin23A/*Rf4^tRf4^t* (*WA352c*/ H4H4*Rf4^tRf4^t*) and Jin23A/*Rf4^tRf4^t*×ZSRf4I (*WA352c*/H4H7*Rf4^t-*) was about twice as high as that in Jin23A/*Rf4^t-* (*WA352c*/H4H4*Rf4^t-*) in qRT-PCR assays (Fig. 5b and Supplementary Fig. 6). Consistent with this, the pattern of *WA352c* transcript levels was opposite to that of *Rf4* (Fig. 5c and Supplementary Fig. 6c).

Together, these results supported the hypothesis that the dosage effect caused by different copy number of functional *Rf4* plays an important role in fertility restoration of CMS-WA.

## The haplotypes with two-copy functional *Rf4* are widely used in three-line hybrid rice breeding

The ability to restore fertility of CMS lines restricts hybrid rice production; therefore, genetic resources with two-copy *Rf4* might be beneficial for breeding stronger restorer lines for CMS-WA in hybrid rice production. To investigate the relationship between haplotypes of the *Rf4* locus and the application of major restorer lines in China, we obtained data about the planting areas of hybrid rice varieties and crossing combinations of restorer lines for CMS-WA from the China Rice Data Center (https://www.ricedata.cn/). With regards to planting area of three-line hybrid rice cultivars, we selected the top sixteen related restorer lines for analysis: six lines (MH63, Ce64-7, CDR22, FuHui838, MH86, and ChengHui727) carried H1 (*Rf4a^M*-*Rf4b^M*), eight lines (MiYang46, Gui99, IR24, R402, Shuhui527, Huazhan, Guanghui998, and Minhui3301) harbored H7 (*Rf4a^l*-*rf4b*), XianHui207 contained H8 (*rf4a*-*Rf4b^M*), and MianHui725 possessed H6 (*Rf4b^M*)

(Supplementary Table 3). As expected, hybrid rice varieties using six restorer lines carrying two-copy *Rf4* (H1) had a larger total planting area (135,998,667 hectares) than the hybrid rice varieties using ten restorer lines having the one-copy *Rf4* (78,015,332 hectares) (Fig. 6). Notably, the two most widely planted hybrid rice varieties were bred from two elite restorer lines, MH63 and Ce64-7, both carrying the two-copy *Rf4* (Supplementary Table 3).

## The *rf4i* variant is dominantly used for the breeding of CMS-WA lines

The rice mitochondrial CMS-WA gene *WA352c* was generated in *O. rufipogon* via multiple rounds of recombination/protogene formation/ functionalization, and *WA352c* has been widely utilized in hybrid rice breeding[1]. However, how *WA352c* co-evolved with *Rf4* remains to be uncovered. To explore the evolutionary relationship of *WA352c* with *Rf4*, we analyzed their sequence structures in different *O. rufipogon* species and rice cultivars. The functional *WA352c* gene only coexisted with three haplotypes (H7, H14, and H28) of the *Rf4* locus in *O. rufipogon* populations (Table 1). This finding suggests that the first CMS-WA germplasm with abortive pollen discovered from an *O. rufipogon* population, called Wild Abortive, carrying a *rf4a-like* variant in addition to *WA352c* (Fig. 7, Table 1).

During the process of CMS-WA line breeding, the *rf4a-like* variant was replaced by *rf4i* through backcrossing with *indica* maintainer lines that harbor the *rf4i* variant, resulting in the current CMS-WA lines (Fig. 7, Table 1). CMS-WA (*WA352c*/*rf4irf4i*), maintainer (*rf4i*), and restorer lines (*Rf4*) made up the CMS-WA/*Rf* system for three-line hybrid rice production (Fig. 7). Based on the above results and the previous finding that *WA352c* originates in *O. rufipogon*[1], it appears that *Rf4* and *rf4* (except for *rf4a* and *rf4aus*) originated earlier than *WA352c* (Fig. 7, Table 1, and Supplementary Data 3) and that the replacement of *rf4i* derived from *indica* type maintainer lines occurred during the creation of modern CMS-WA lines.

## PCR-based molecular markers for genotyping the *Rf4* locus to assist hybrid rice breeding

To facilitate the identification of *Rf4* (*rf4*) haplotypes in hybrid rice breeding, we selected and optimized a set of eight *Rf4* variant-specific PCR-based markers based on the SNPs at the *Rf4* locus (Supplementary Table 1). To confirm the utility of this set of markers, these primers were used to investigate the genotypes of 304 Asian cultivated rice germplasms. PCR products of these lines were first divided into three types, Copy-a, Copy-b, and Copy-a/-b, using two PCR markers (Copy-a-332 bp-F/R and Copy-b-282 bp-F/R), which generated 332-bp- and 282-bp PCR products from Copy-a and Copy-b, respectively (Fig. 8 and Supplementary Table 1). Then the variants and haplotypes of *Rf4* and *rf4* were determined using six specific primer sets. The fragments amplified from one-copy *Rf4* from the Copy-a or Copy-b in rice varieties with the *Rf4a^l* haplotype (such as Jalmagna and GH102) or *Rf4b^M* haplotype (such as MH725 and R60) were 262 bp long (Fig. 8a). No products of functional *Rf4* were amplified from *japonica*, *indica*, and *aus* varieties carrying haplotypes of *rf4j*, *rf4i*, and *rf4aus*, but a 372-bp, 358-bp, and 351-bp products of *rf4* were obtained from the Copy-a of these varieties, respectively (Fig. 8b).

Varieties with the *rf4a-Rf4b^M* haplotype (such as GH993 and XH207) or the *Rf4a^l-rf4b* haplotype (such as ZSRf4I and IR8) all carried one-copy *Rf4* (Fig. 8a). They shared four common PCR products: Copy-a (332 bp), Copy-b (282 bp), *Rf4* (262 bp), and *rf4a/b* (197 bp). Varieties with the *rf4a-Rf4b^M* haplotype also generated another product: *rf4a* (446 bp). Varieties carrying two-copy *Rf4* with the *Rf4a^M-Rf4b^M* haplotype (such as MH63, SH498, FH838, and IR30) generated three PCR products: *Rf4* (262 bp), Copy-a (332 bp), and Copy-b (282 bp) (Fig. 8c). Taken together, these results demonstrated that these primer sets are useful PCR markers for the rapid genotyping to accelerate the screening of strong restorer lines with the two-copy *Rf4*.

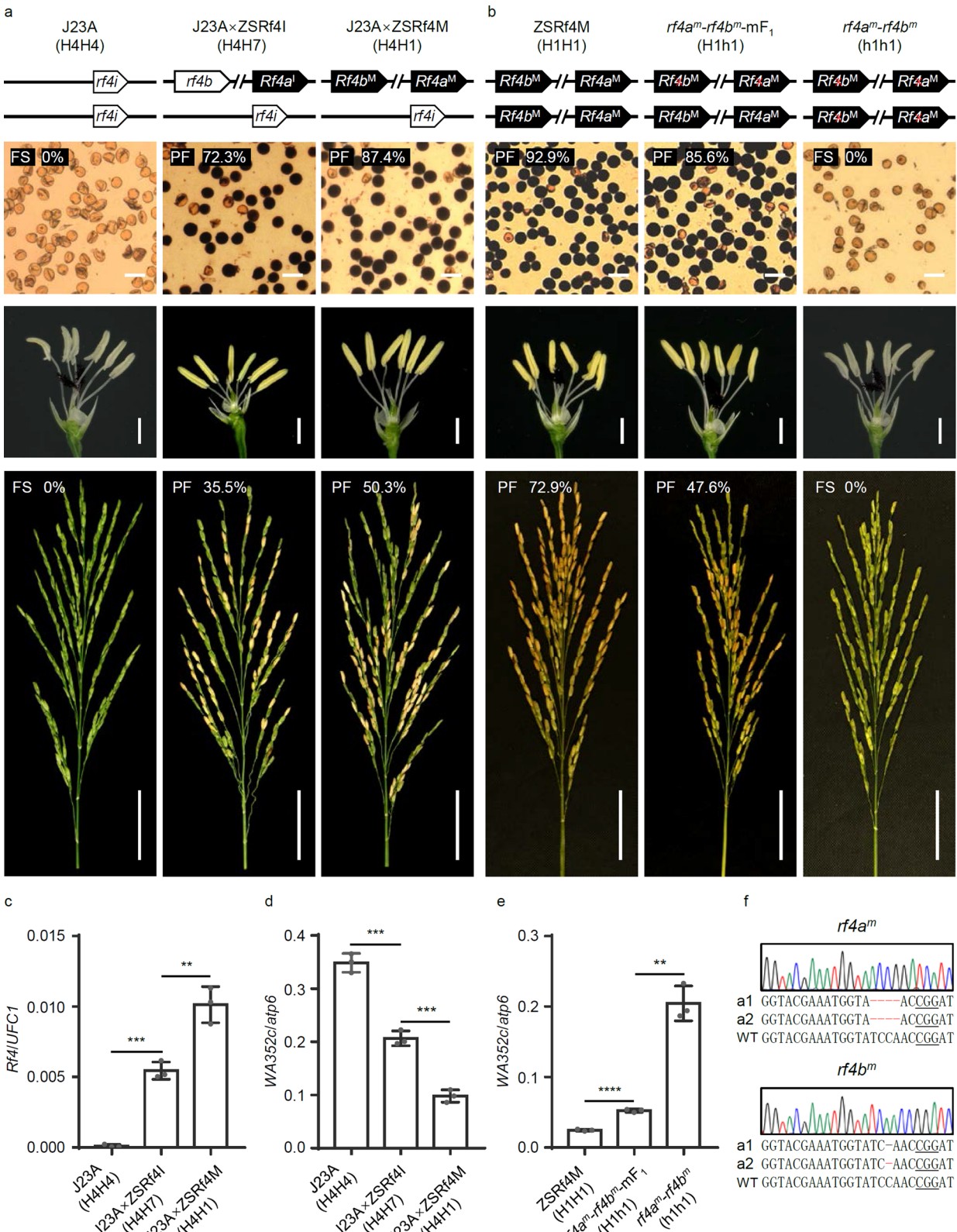

**Fig. 4 | Validation of the dosage effect of *Rf4* using *Rf4* near-isogenic lines.**
**a**, **b** Pollen viability based on staining (upper panels), anther phenotype (middle panels), and seed setting rate (lower panels) of Jin23A×ZSRf4I (*WA352c*/H4H7), Jin23A×ZSRf4M (*WA352c*/H4H1) and *rf4aᵐ-rf4bᵐ-mF₁/rf4aᵐ-rf4bᵐ* mutant lines (by CRISPR/Cas9 editing) in the ZSRf4M background. Red "×" indicates the non-functional *rf4a/bᵐ* after knockout of *Rf4a/bᴹ*. Scale bars: 50 μm in the upper panels, 1 cm in the middle panels, and 5 cm in the lower panels. **c**–**e** Transcript levels of *Rf4* (**c**) and *WA352c* (**d**, **e**) in different lines. *UFC1* (*UFM1-Conjugating Enzyme 1*) and *atp6*

(a mitochondrial gene) served as internal references for *Rf4* and *WA352c* expression, respectively. Data are shown as mean ± SD, *n* = 3 biological replicates. Significant differences between two samples were determined by two-tailed Student's *t*-test (**P* < 0.01, ***P* < 0.001, ****P* < 0.0001). **f** Sequencing of the *Rf4a/bᴹ*-knockout plants derived from CRISPR/Cas9 editing. The underlined bases show protospacer adjacent motifs (PAMs). The positions highlighted in red indicate the targeted mutations. Source data are provided as a Source data file.

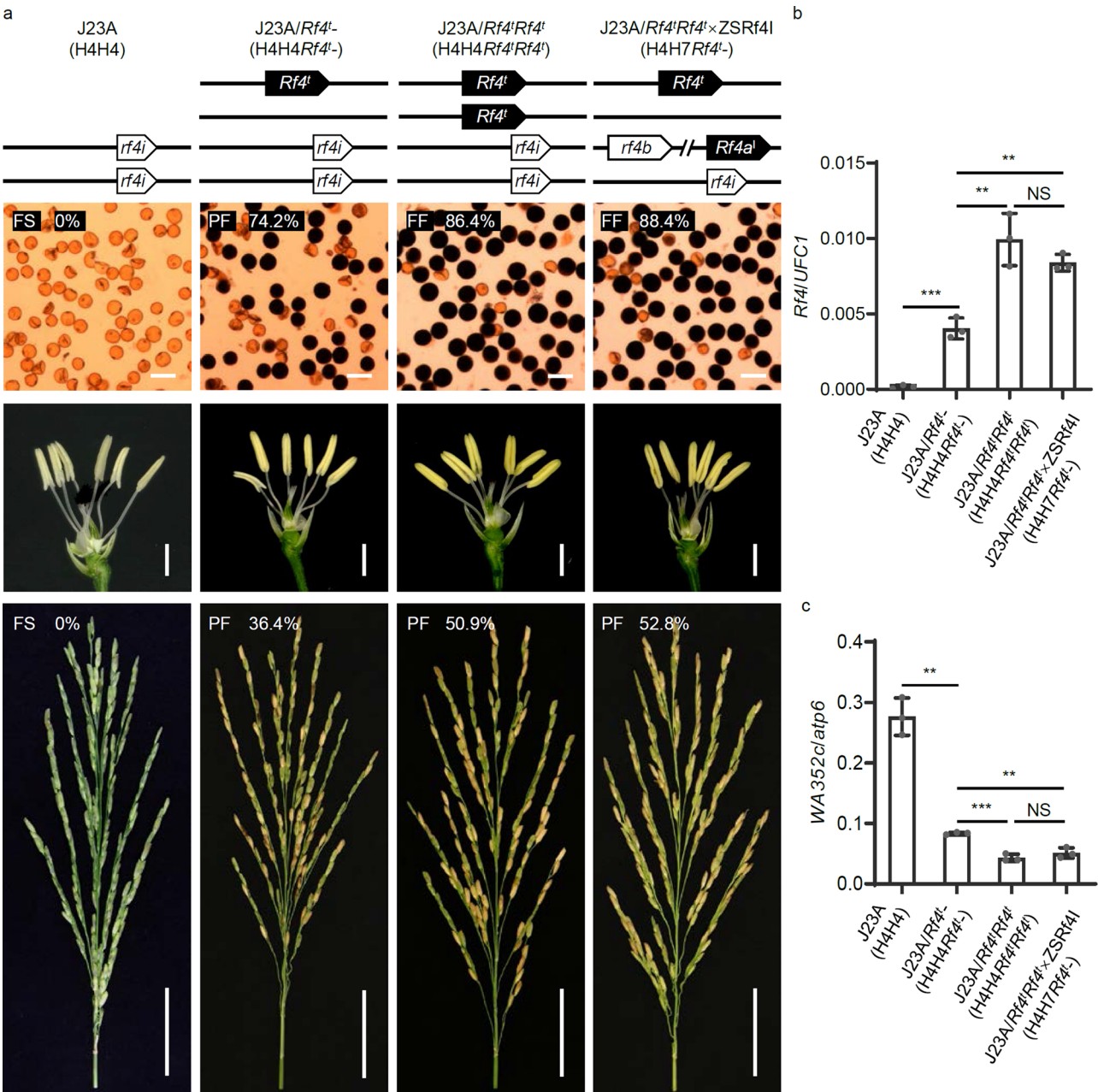

**Fig. 5 | Confirmation of the dosage effect of *Rf4* in complemention lines. a** Pollen staining (upper panels) to reveal viable pollen (dark), anther phenotype (middle panels) and seed setting rate (lower panels) of Jin23A/*Rf4ᵗ*- (*WA352c*/H4H4*Rf4ᵗ*-), Jin23A/*Rf4ᵗRf4ᵗ* (*WA352c*/H4H4*Rf4ᵗRf4ᵗ*), and Jin23A/*Rf4ᵗRf4ᵗ*×ZSRf4I (*WA352c*/H4H7*Rf4ᵗ*-). Scale bars: 50 µm in the upper panels, 1 cm in the middle panels, and 5 cm in the lower panels. **b, c** Transcript levels of *Rf4* (**b**) and *WA352c* (**c**) in different

lines. "*Rf4ᵗ*" indicates the *Rf4* transgene, "*Rf4ᵗ*-" indicates transgenic hemizygotes. Data are shown as mean ± SD, *n* = 3 biological replicates. Significant differences between two samples were determined by two-tailed Student's *t*-test (**P < 0.01, ***P < 0.001, and NS represents No Significance). Source data are provided as a Source data file.

## Discussion

The CMS/*Rf* system has been widely used in hybrid seed production[41,42]. Functional CMS genes originated and evolved via a multiple recombination, protogene formation, and complex diversification[1,43]. However, our understanding of *Rf* gene evolution and its co-evolution with CMS genes is very limited. Here, we demonstrated the evolutionary plasticity of *Rf4* in terms of sequence structure, copy number, and biological function during the evolution of the *Oryza* genus.

Structural variation (duplications, deletions, insertions, and chromosomal rearrangements) is an important driver of phenotypic variability in crop traits during the processes of evolution and domestication[28,44,45]. Based on public genome information for rice and sequencing of PCR products, we reconstructed a diagram of the genetic structure of the *Rf4* complex locus in three rice varieties (Fig. 1a) and its genetic structure in different rice cultivars. We discovered that the *Rf4* locus shows CNV associated with the restoration effect of this locus. In total, eight haplotypes were identified in modern rice cultivars, including the two-copy haplotypes such as H1 (*Rf4aᴹ*-*Rf4bᴹ*), H7 (*Rf4aᴵ*-*rf4b*), and H8 (*rf4a*-*Rf4bᴹ*), as well as the one-copy haplotypes such as H2 (*Rf4aᴵ*), H3 (*rf4aus*), H4 (*rf4i*), H5 (*rf4j*), and H6 (*Rf4bᴹ*) (Fig. 1b). The functional *Rf4* variants enabled the degradation of *WA352c* transcripts through post-transcriptional mechanism and restore the male fertility of CMS-WA, whereas *rf4a*, *rf4b*, *rf4aus*, and *rf4j*

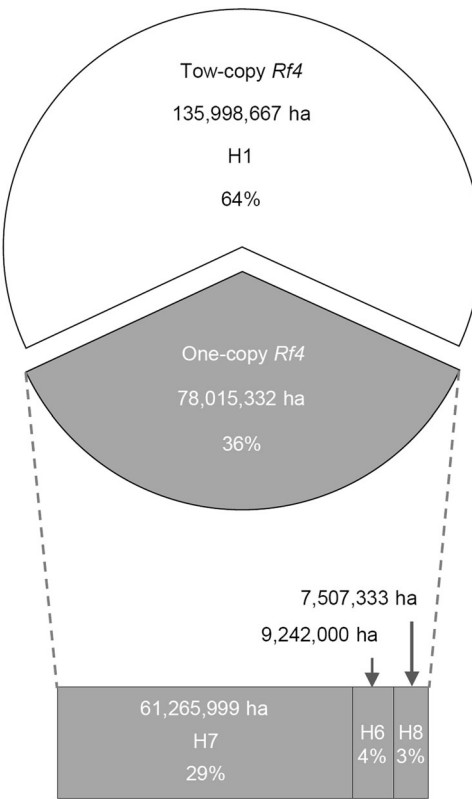

**Fig. 6 | Usage of *Rf4* haplotypes in major restorer lines for hybrid rice production in China.** Total planting areas and relative percentage of hybrid rice varieties in China using 6 restorer lines carrying the two-copy *Rf4* (H1) and 10 restorer lines with one-copy *Rf4* (H6, H7, H8). The data information is given in Supplementary Table 3.

are non-functional variants like *rf4i* (Fig. 3b and Supplementary Fig. 4). Furthermore, our findings indicated that the functionalization of Rf4 protein is affected by 14 amino acids in three motifs from PPR13 to PPR15 (Fig. 3). How these amino acids determine the fertility restoration activity of the protein requires further study.

*Rf* genes identified in plants mainly encode a subclass of PPR proteins that are subjected to rapid evolution[20,46]. The rice *Rf* loci encoding PPR proteins show considerable structural and sequence variation within and among *Oryza* species[22,47]. In this study, we revealed that the ancestral *Rf4*-like sequence first appears at the "Copy-a" site of the *Rf4* locus in *O. meyeriana* (GG-genome), which is regarded as the oldest wild rice species. Subsequently, the *Rf4* locus likely experienced more expansions in the genome of multiple wild rice species, leading to the sequence variation and duplication of *Rf4* or *rf4* in the Copy-a and/or Copy-b sites (Fig. 2 and Supplementary Data 3).

**Table 1 | Association of *WA352c* and *Rf4* haplotypes in different populations of *O. rufipogon* and *O. sativa***

| Species | Haplotype | CMS gene |
|---|---|---|
| *O. rufipogon* | H7 (*Rf4aᴵ-rf4b*) | *WA352c* |
| | H14 (*rf4a-like*) | *WA352c* |
| | H28 (*Rf4aᴹ-rf4b-like*) | *WA352c* |
| *O. sativa* | H7 (*Rf4aᴵ-rf4b*) | *WA352c* |
| | H8 (*rf4a-Rf4bᴹ*) | *WA352c* |
| | H1 (*Rf4aᴹ-Rf4bᴹ*) | *WA352c* |
| | H6 (*Rf4bᴹ*) | *WA352c* |
| | H4 (*rf4i*) | *WA352c* |

One-copy haplotypes of *Rf4* might have appeared early in the evolution of the *Oryza* genus, and different haplotypes of *Rf4* might have multiple origins and co-exist in the wild rice gene pool. Based on the 68 variants identified from the Copy-a and/or Copy-b sites, and 69 haplotypes at the *Rf4* locus, we reasoned that the *Rf4* locus shows high plasticity in *Oryza* (Fig. 2 and Supplementary Fig. 2).

During evolution, along with natural and human selections, the nascent one-copy and two-copy *Rf4* haplotypes gradually migrated into the lineages of *O. rufipogon*, *O. nivara*, and *O. sativa*, and finally became enriched in modern Asian cultivars. However, the *Rf4* gene flow might have failed to pass through certain bottleneck events to enter the lineages of *O. longistaminata*, *O. barthii*, and *O. glaberrima*, so the *Rf4* haplotypes were not detected in the tested accessions. Alternatively, considerable structural and sequence variations might occur in the lineages of *O. longistaminata*, *O. barthii*, and *O. glaberrima*, leading to gene loss in the *Rf4* locus (Fig. 2).

Notably, there are only eight *Rf4* haplotypes (H1–H8) enriched in modern rice cultivars, and the two-copy *Rf4* haplotypes (H1) are predominant in three-line hybrid rice varieties (Fig. 6). Among the eight *Rf4* haplotypes, five are widely distributed in the wild rice accessions, suggesting that these haplotypes might have preexisted in different wild rice species (Fig. 2 and Supplementary Data 3). Since the *Rf4aᴵ* variant is only detected in the one-copy H2 (*Rf4aᴵ*) in modern cultivars and the two-copy H7 (*Rf4aᴵ-rf4b*) in wild rice species; the H2 might have directly originated from H7 due to loss of *rf4b* during domestication (Fig. 2).

Gene dosage effects caused by CNV are always accompanied by changes in gene expression levels and protein accumulation. These processes are involved in human diseases and agricultural traits in both animals and plants[36,48]. We found that the *Rf4* complex locus had different forms of CNV, which resulted in a gene dosage effect. Varieties or hybrids carrying two/four copies of *Rf4* showed higher pollen fertility, full anthers, and higher seed setting rates than those carrying one/two copies of *Rf4* (Figs. 4a, b, 5a and Supplementary Figs. 5a, b, 6a). By contrast, the level of *WA352c* transcript decreased with increasing copy of *Rf4* (Figs. 4c–e, 5b, c and Supplementary Figs. 5c–e, 6b, c). We also investigated the application of *Rf4* haplotypes in three-line hybrid rice production in China. As expected, restorer lines containing two-copy *Rf4* (such as MH63, FH838, etc.) had larger planting areas than restorer lines carrying one-copy *Rf4* (such as MiYang46, Gui99, etc.) (Fig. 6 and Supplementary Table 3), suggesting that stronger capacity for fertility restoration is an important trait selected by breeders. Therefore, rice materials carrying the two-copy *Rf4* (*Rf4aᴹ-Rf4bᴹ*) were preferentially selected by breeders to generate strong restorer lines for three-line hybrid rice production. Notably, *Rf4* is located in a hotspot of a *Rf* gene cluster on Chromosome 10 in rice, and clusters with other restorer genes including *Rf1a/Rf5* for CMS-BT/CMS-HL, *Rf1b* for CMS-BT[24,25], and *Rf19* for CMS-FA[26]. Pyramiding multiple restorer genes in a restorer line can expand the crossbreeding combinations and increase planting area of related hybrid varieties.

In general, *Rf* or *Rf-like* genes show certain characteristic features, including genomic clustering and unique evolutionary patterns[19,20,22]. Functional *Rf4* variants exist in ancient wild rice species (EE- and CC-genome), landrace, and modern cultivars (Fig. 2, Supplementary Fig. 2, and Supplementary Data 2–4), pointing to its evolutionary conservation. Since *WA352c* is a newly evolved mitochondrial sterility gene in *O. rufipogon*[1], *Rf4* (and its variants) may have had other undiscovered intrinsic functions prior to its neofunctionalization as a fertility restorer for *WA352c* during the adaptive evolution of wild rice. Moreover, we determined that all current CMS-WA lines used for hybrid rice production contain the combination of *WA352c* and *rf4i*, which is not present in *O. rufipogon* (Table 1 and Supplementary Data 3). H4 (*rf4i*) is a pseudogene due to the insertion of fragments leading to premature termination[15], which might firstly occur in *O. australiensis* and might have become predominant in rice landraces (Fig. 2 and Supplementary

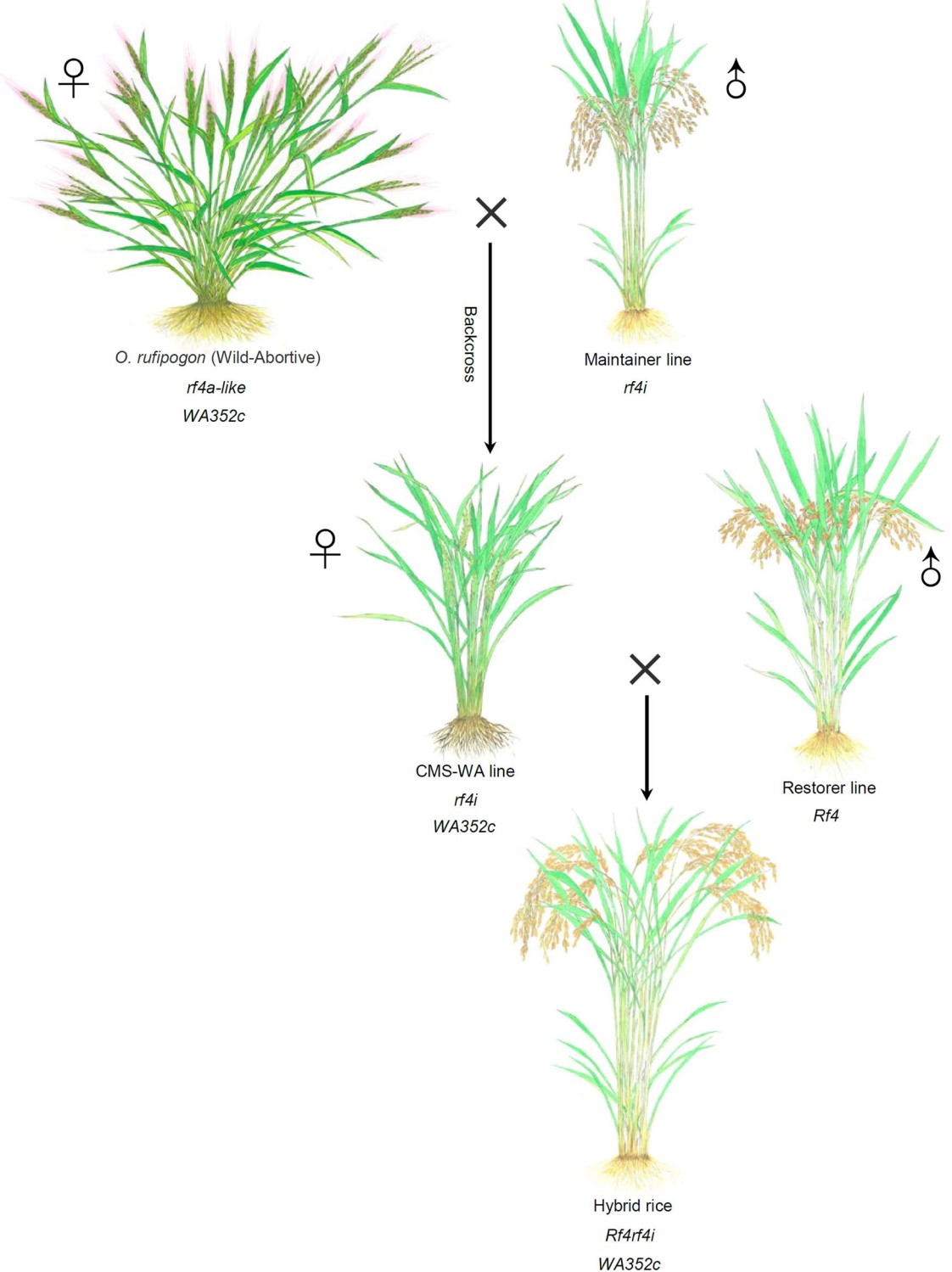

**Fig. 7 | A speculative model for formation of the CMS-WA/Rf system.** An *Oryza rufipogon* population (Wild Abortive) with the mitochondrial sterility gene *WA352c* and non-functional *rf4a-like* showing pollen abortion was found and used as the female parent for breeding CMS-WA lines by backcrossing with *indica* maintainer lines containing *rf4i*. A hybrid rice variety was bred by crossing the CMS-WA line with a restorer line (carrying one-copy or two-copy *Rf4*).

Data 3 and 4). Based on our findings, we suggest that the *rf4a-like* variant (H14) and *WA352c* is the original combination in *O. rufipogon*, which was later termed Wild-Abortive germplasm (Fig. 7, Table 1, and Supplementary Data 3). In early 1970s in China, this CMS germplasm was explored for breeding CMS-WA lines by backcrossing with recurrent parental (*indica*) varieties (maintainer lines) carrying the *rf4i* variant (Fig. 7).

Molecular marker-assisted selection (MAS) is an effective tool for accelerating the screening of crops with desired genotypes[49,50]. In this study, we developed a set of precise PCR-based molecular markers to identify *Rf4* and *rf4* haplotypes at the *Rf4* locus. The functional *Rf4* variants in a variety could be identified using the *Rf4* marker, and CNV of the *Rf4* locus could be characterized using four markers: Copy-a, Copy-a, *rf4a*, and *rf4a/b* (Fig. 8a, b). Thus, we recommend that

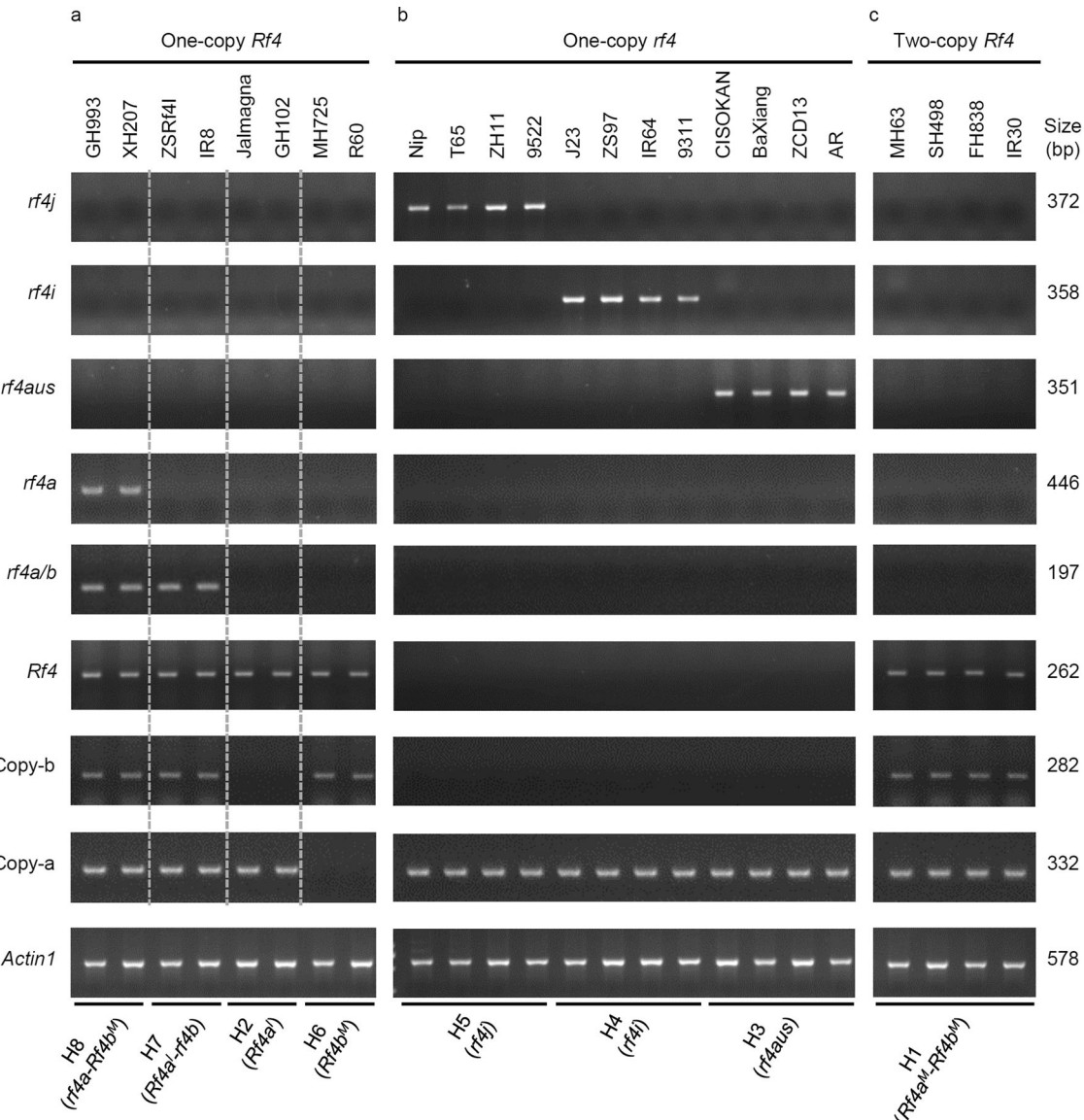

**Fig. 8 | Identifying different *Rf4* haplotypes in cultivated rice.** Eight pairs of variant-specific primers were used for PCR to determine the *Rf4* and *rf4* haplotypes of different rice lines. **a** One-copy *Rf4*, **b** one-copy *rf4*, **c** two-copy *Rf4*. White dashed lines separate four different patterns of one-copy *Rf4*. GH993 (Guanghui993), XH207 (Xianhui207), ZSRf4I (Zhenshan Rf4I), IR8, Jalmagna, GH102 (Guanghui 102), MH725 (Mianhui725), R60, J23 (Jin23), ZS97 (Zhenshan97), IR64, 9311, MH63

(Minghui63), SH498 (Shuhui498), FH838 (Fuhui838), and IR30 are *indica* cultivars. Nip, T65, ZH11, and 9522 are *japonica* cultivars. CISOKAN, BaXiang, AR (Albania Rice), and ZCD13 (Zacaodao13) are circum-*aus* cultivars. *Actin1* was used as the PCR control. The PCR experiments for each sample were independently repeated at least three times with similar results. Source data are provided as a Source data file.

breeders use five markers (*Rf4*, Copy-a, Copy-b, *rf4a*, and *rf4a/b*) to identify germplasm with two-copy *Rf4* to generate elite restorer lines (Fig. 8c). A one-copy *rf4* haplotype is present in Copy-a, the *japonica* and *aus* varieties harbor the *rf4j* and *rf4aus* variants, respectively, while all CMS-WA lines and maintainer lines examined harbor the *rf4i* variant (Fig. 8b). Therefore, the *rf4i* marker is an accurate marker for MAS of new CMS-WA lines and maintainer lines in landrace and/or cultivated rice backgrounds. Rice germplasms carrying the *rf4aus* and *rf4j* haplotypes lack the function for fertility restoration (Fig. 3b), but they have the potential to be new CMS/maintainer lines for future hybrid rice breeding.

Meanwhile, landraces and modern Asian rice varieties that carry *Rf4* variants (including the one-copy *Rf4* and two-copy *Rf4* ones) were screened as restorer lines by human selection (Figs. 2, 7). Although breeders may not be aware of the CNV of *Rf4* in the breeding programs, they selected successful restorer lines based on their ability of fertility

restoration along with other beneficial agronomic traits. We believe that our findings will benefit the high-efficiency breeding of CMS/ maintainer lines and restorer lines to improve hybrid rice yields in the future.

## Methods

### Plant materials
All plant materials were grown in the rice field in South China Agricultural University, Guangzhou, China. The wild rice accessions were selected from "Ding's Rice Collection" in South China Agricultural University. Landraces and cultivars of *O. sativa* were collected from different provinces in China and used to examine the haplotypes of *Rf4* and *rf4*. The CMS-WA line Jin23A served as the recipient for the *Rf4* and *rf4* functional complementation assays. Near-isogenic lines ZSRf4I and ZSRf4M were derived from backcrosses of the CMS-WA line ZS97A with the restorer line IR24 and MH63, respectively.

## Vector construction and plant transformation

The coding sequences of *Rf4^M*, *rf4a*, *rf4b*, *rf4aus*, and *rf4j* driven by the native promoter of *Rf4^M* and followed by a Nos terminator were cloned into modified pCAMBIA1300 binary vectors by the Gibson assembly method[51]. The CRISPR/Cas9 knockout plasmid for *Rf4* was constructed as previously reported[52]. Briefly, target sequence (ggtacgaaatggtatccaac) selected in the coding region of *Rf4* were introduced into sgRNA expression cassette by overlapping PCR, then the cassette was cloned into the CRISPR/Cas9 binary vector by the Gibson assembly method[52]. These generated constructs were introduced into *Agrobacterium tumefaciens* strain EHA105, and then transformed into Jin23A or ZSRf4M by the *Agrobacterium tumefaciens*-mediated method. Primers for vector construction are listed in Supplementary Table 1.

## Genotyping and phenotypic characterization of fertility in the transgenic lines

Pollen grains were stained with 1% $I_2$-KI solution and imaged under a light microscope (Axio Observer D1, Carl Zeiss, Oberkochen, Germany) to measure the pollen viability rate, and the main panicle was collected from each individual to examine the seed setting rate and photographed with a camera (Canon, Japan). We obtained at least 10 independent transgenic lines for each complementary construct with similar phenotype and showed the phenotype of three independent lines in Fig. 3b and Supplementary Fig. 4. The flowers and main panicles from ten different plants of each independent line were selected for statistical analysis of pollen viability and seed setting rate, respectively. The T-DNA flanking sequences of *Rf4* were identified using mhiTAIL-PCR[53]. T-DNA tags of the transgenic plants were genotyped using specific primer sets. In addition, the genotypes of *Rf4*-variants were determined using the *Rf4*-linked marker. Four independent *Rf4*-knockout plants were genotyped through PCR-based sequencing. Primers used for genotyping complementary or knockout lines are listed in Supplementary Table 1.

## PCR-based haplotype analysis

Molecular markers (Supplementary Table 1) were designed based on sequence differences among the Copy-a, Copy-b, *Rf4*, *rf4a*, *rf4a/b*, *rf4i*, *rf4aus*, and *rf4j* to identify the haplotypes of *Rf4* and *rf4* in cultivated rice by PCR. Each 20 μL reaction contained 10 μL 2× Taq PCR StarMix with Loading Dye (Genstar, China), 0.125–0.25 mM primer, and 1 μL DNA (50–100 ng). PCR was performed under the following conditions: 94 °C for 4 min; 29 cycles of 94 °C for 30 s, 59 °C for 30 s, and 72 °C for 20 s; a final extension of 2 min at 72 °C. Long DNA fragments of the Copy-a and Copy-b were amplified with specific primers using STI PCR[54]. The amplified DNA fragments were sequenced and analyzed by BLAST (http://www.ncbi.nlm.nih.gov/BLAST/) and Clustal Omega (https://www.ebi.ac.uk/Tools/msa/clustalo/). Sequencing of *WA352c* was performed using previously described primers[15]. All the PCR experiments for each sample were performed with at least three replicates. The PCR products were separated on 1.5–2% agarose gels by conventional electrophoresis.

## RNA extraction and qRT-PCR analysis

Total RNA was extracted from rice anthers at the microspore mother cell stage[55] using TRIzol reagent (Ambion, USA), followed by reverse transcription into cDNA using a HiScript III 1st Strand cDNA Synthesis Kit (Vazyme, China) with oligo(dT) and random hexamers according to the manufacturer's instructions. qRT-PCR analysis was performed on the Bio-Rad CFX Real-Time PCR system (Bio-Rad, USA). Expression analysis of *Rf4* and *WA352c* was performed by qRT-PCR with specific primers (Supplementary Table 1) with three biological replicates; *UFC1* (encoding a homolog of UFM1-Conjugating Enzyme 1, GenBank number: AK059551), which is a constitutive expression gene with medium level suitable as internal reference gene[56], and *atp6* (the mitochondrial gene encoding a subunit of ATP synthase)[15],

were used as internal controls for the expression of *Rf4* and *WA352c*, respectively.

## Phylogenetic and evolutionary analyses of the *Rf4* locus

The orthologs of *Rf4* in the Poaceae family were identified for phylogenetic analysis of *Rf4* using the tblastn program in the GenBank database (https://www.ncbi.nlm.nih.gov/) based on the *Rf4* sequence in the cultivars. To trace the divergence and perform phylogenetic analysis of the *Rf4* locus, *Rf4* and *rf4* fragments were amplified and sequenced in all wild rice, landrace, and cultivated rice accessions with specific primers (Supplementary Table 1). The nucleotide sequences of *Rf4* homologs were aligned with MAFFT v.7.475[57] and manually adjusted using MEGA X v.10.1.7[58]. Phylogenetic analysis was conducted using the maximum likelihood method with MEGA X v.10.1.7. Bootstrapping with 1000 replicates was used to generate the best tree and its branches. The phylogenetic trees were then plotted with the online tool Interactive Tree of Life (https://itol.embl.de/).

## Reporting summary

Further information on research design is available in the Nature Portfolio Reporting Summary linked to this article.

## Data availability

All data supporting the conclusions of this work are present within the paper and its Supplementary Information files. Source data are provided with this paper.

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

## Acknowledgements

We thank Professors Xiangdong Liu from South China Agricultural University, Qian Qian and Jianlong Xu from Institute of Crop Sciences, Chinese Academy of Agricultural Sciences for providing some rice germplasms. This work was supported by the Open Competition Program of the Top Ten Critical Priorities of Agricultural Science and

Technology Innovation for the 14th Five-Year Plan of Guangdong Province (No. 2022SDZG05 to L.C.), the National Natural Science Foundation of China (No. 32030080 to L.C.; No. 32101695 to Z.Z.), the Guangdong Natural Science Funds for Distinguished Young Scholars (No. 2021B1515020089 to Y.X.) and the Key Research Program of the Guangzhou Science, Technology and Innovation Commission (No. 201904020030 to Y.-G.L.).

## Author contributions

L.C., Z.Z., and Y.X. designed research; Z.Z and D.Z. performed most of the experiments; J.H., H.M., Z. Zhang, X.G., H.T., X.X., and J.P. performed some of the experiments; Y.-G.L. and F.X. provided theoretical guidance on the experiments and revised the manuscript; L.C., Z.Z., and Y.X. wrote the paper.

## Competing interests

The authors declare no competing interests.
