## [Peer Review File · Nature Communications]

Copy number variation of the restorer Rf4 underlies human selection of three-line hybrid rice breedingReviewers' Comments:

Reviewer #1:

Remarks to the Author:

Hybrid rice make great contributions for ensuring the food security of the world, especially in China. CMS-WA type hybrid rice is first found and applied in China, and account for more than 80% three-line hybrid rice. Here, authors detailed analyzed the Rf4 region in wild rice and cultivated rice, and raised the hypothesis that Rf4 locus emerged earlier than the CMS-WA gene WA352. They cloned these eight haplotypes of the Rf4 locus and verified that the two-copy Rf4 haplotype has stronger fertility restoration ability compared with single-copy haplotypes. The conclusion is that dosage effect of Rf4 is quite important for breeding the elite restorer lines. This research will increase our understanding of CMS/Rf systems and benefit crop breeding for utilization of heterosis in CMS-WA system. I strongly recommended for publication.

Minor questions:

Did the authors analyzed the distribution of the wild rice, landraces? Maybe it is important to explain the origin of WA-352 and Rf4.

In figure 5c, what is UFC1? Please clarified it in Materials and Methods in detail.

In supplemental figure 5e, the maximum of ordinate should be 100. I suggest it can be revised as 5a and 5c

Reviewer #2:

Remarks to the Author:

Key results

This is a report of rice Rf4 that restores pollen fertility to CW-CMS plant. Variants of rice Rf4 are considered as haplotypes consisting of one to several PPR protein-coding genes. The strength of Rf4 haplotypes to restore pollen fertility is explained by gene dose effect of the specific copy of the PPR gene. Strong haplotype is favored in rice hybrid varieties.

Validity

I understand the high plasticity of Rf4. But the manuscript also claims purified selection acting on Rf4. How are these be reconciled?

Significance

This is the first report of gene dose effect of rice Rf. But not in crop. Therefore, line 92 is invalid. Gene dose effect of Rf has been reported in <https://doi.org/10.1007/s00122-018-3211-6>. The molecular basis of gene dose effect of Rf has been reported in <https://doi.org/10.1186/s12870-020-02721-9>. Relationship between Rf variant and plant breeding has been reported in <https://doi.org/10.1007/s11032-013-9854-8>.

Analytical approach

On the analysis of dN/dS ratio, care should be needed. Especially selecting samples.

Suggested improvements

The manuscript is inflated by unnecessary data, and streamlining the manuscript could make it better.

Clarity and context

The manuscript is poor in presentation. It is difficult to follow the meaning. This is in part due to the ambiguous usage of terms. For example, genetic symbol is not consistent. I see rf3 instead of rf4 (line 225). Is rf3 an allele of rf4? I think it is a kind of laboratory jargon. Please follow the rule of genetics. I also see confusion between 'allele' and 'haplotype' (e.g. line 176; this is just an example, and check throughout the manuscript). A number of symbols are used without definition. I cannot understand

the content of lines 161 to 205 in part because of no instruction which 'allele' corresponds to the homologue in Fig. 3.

References

Kato et al. (<https://doi.org/10.1038/sj.hdy.6801026>) should be cited for the structural variation of rice Rf-1 on the chromosome 10 because this is the first to report the polymorphism of this locus.

Reviewer #3:

The paper is a follow-up work of the classical CMS gene WA352 controlling rice WA CMS. Based on former studies of this CMS system, the author tried to further explore the evolutionary trajectory/dynamics of the CMS restorer gene Rf. The wet experiments are solid and show significant potential in improving the efficiency of this pollen control system. However, I am mainly concerned that the evidence of parallel evolution of Rf4 shown in this study is not detailed and clear. The reliable evidence to show parallel evolution is that those species were under some specific biotic or abiotic selection pressure and gradually evolved similar phenotypes. For example, parallel evolution was typically reported in *Arabidopsis*, *Gasterosteus aculeatus*, and flying frog under which the selection pressure probably is climatic factors, freshwater/saltwater niches, and predation pressure, respectively. However, the author did not discuss the potential driver of promoting the parallel evolution of Rf as well.

The author composed some figures to show the correspondence relationship between the haplotypes and wild species/varieties. They avoided choosing the phylogenetic trees of those species to show the variance pattern of CNV and SV, making the evidence not explicit. Additionally, in Supplementary Table 3 and Supplementary Table 4, the author missed the information on the geographical distribution of those Landrace/Wild rice accessions and the main planting region of those cultivated rice.

Additionally, many narratives of language should be largely improved, such as “have important uses”, “robust fertility restorer (Rf) genes”, “first major Rf”, “by natural and artificial selection during domestication.”.

Comments:

Line1: The “dosage” mentioned in the title seems ambiguous. The author may need to clarify whether it is transcriptional dosage or translational dosage based on their expression experiments, or just simply refer to two copies of it.

Line15-16: What do the authors really mean by “robust fertility restorer (Rf) genes”? Please make a clarification. “first major Rf for Wild-Abortive CMS,” what does “major” exactly mean? The gene frequency is highest or the ability to restore fertility is the strongest. Cite related paper at the end of line16.

Line17-18: “However, the formation of the CMS-WA/Rf system and Rf4 evolution during domestication remain elusive.” The word “formation” coordinates well with “during domestication”; however, “Rf4 evolution” does not coordinate with “during domestication” since evolution emphasizes a natural process but not domestication by humans. The author should think about the difference between “domestication” and “artificial selection.” I prefer to use “artificial selection” here than “domestication” since humans would not expect to domesticate a male sterile phenotype.

Line29: Please remove “for increased yield” unless you have preliminary evidence in this paper.

Line36: The author should cite the pioneer paper of each type of CMS.

Line37-38: Could the author please clarify in which years range and which regions (or countries) CMS-WA occurs in approximately 90% of hybrid rice? “Therefore,” this seems illogical here. The author should revise it since CMS-WA occurs in approximately 90% of hybrid rice. It’s great enough compared to other systems. How the author deduces that “Therefore, understanding the evolution and domestication of related fertility restorer (Rf) genes provides important information for facilitating hybrid rice breeding.” Additionally, the author should check the citation 4 and 5 here for the most reliable resource of this information.

Line 41-43: "has undergone ~15 million years of evolution" could be revised to "has undergone significant diversification over the course of evolution, with many of these species diverging even within the past 1 million years."

Line 44: The citation "6-8" could be replaced with a more relevant reference that supports the statement. For instance, the author could cite the pioneer paper that reveals this view by abundant taxonomy evidence or a paper with most molecular evidence mainly underlying this statement.

Line92: check this paper, “Evolution of the Brassicaceae-specific MS5-Like family and neofunctionalization of the novel MALE STERILITY 5 gene essential for male fertility in Brassica napus”(2020), they reported a restorer gene could inhibit another allele in B. napus in a dosage-dependent manner.

Line111: The author could clarify how they confirmed that these genomes have the most reliable assembly within these Rf regions. The author could indicate the version and cite the papers and resource links.

Line118: The author could indicate whether there is any functional evidence showing that the rf4i gene was pseudogenized due to a premature termination codon.

Line121: “are identical”, please indicate the % of identity and similarity between the two genes. “as are the regions 7.5 kb upstream of the start codon and 1.5 kb downstream of the stop codon”, it’s hard to believe that the 7.5 kb upstream and 1.5 kb downstream are identical. Usually, the duplicate at least experiences genetic drift and have some synonymous SNP.

Line124: “Therefore,”, illogical problem.

Line126: Please indicate the basic information and the reason why chose these 304 cultivars. How could the 304 cultivars figure out all the alleles by the primers based on those SNPs which only come from the four publicly available rice genome information?

Line127: since Line121-122 indicate that “as are the regions 7.5 kb upstream of the start codon and 1.5 kb downstream of the stop codon”, how can the author design the site-specific PCR primers based on SNP?

Line151-152: “these motifs are important for the fertility restoration of CMS-WA”, should be “amino acid substitutions at the PPR13, PPR14, and PPR15 motifs are important for the fertility restoration of CMS-WA”. What are the variances of other PPR motif (PPR1- PPR12..) between those Rf and rf?

Line172: I strongly suggest that the author make an additional Figure showing the aa BLAST among all those putative homologs within Poaceae genomes.

Line174: “identity”, please indicate the % here.

Line692: It’s hard to sort the detail by looking at this circle tree.

Line705: I am concerned with the Fig. 4 either because it’s unclear to show the evolutionary events such as “expansion” and “Hybridization, gene gain and loss” or because it is non-compliant on exhibiting the evolutionary time scale (or no clear evolutionary time of all wild/cultivated rice). For example, the author exhibits three panels in the figure 4 to show the 3 stages: ancestral, wild rice and cultivated rice. I am not sure whether the “expansion” between ancestral and wild rice means more copy in each wild rice species or more different alleles originated. If it’s the former, we didn’t find there is more Rf/rf copy in each wild rice such as rfi/rfj/rfaus than in ancestral, if it’s the latter, then this process is not expansion. Additionally, in “wild rice” panel, which species divergent time do the allele rfa/rfb or the allele Rfa/Rfb originated? I strongly suggest the author to implement an alternative by incorporate the phylogenetic tree into the Figure4 (left panel is phylogenetic tree, right panel is the structure of alleles). BTW, for the Supplementary Fig. 3, the author didn’t choose the two species: *O. barthii* and *O. glaberrima*, which originated between *O. glumaepatula* and the ancestor of *O. niva* and *O. rufi* (Stein, Nature genetics 2018). I would expect that there will probably be more interesting CNV and SV within this branch and it would be helpful for the author trace Rf. The author could just simply check the genome sequence of Rf region.

Line188: “in the *O. meyeriana* species (GG-genome, ancestor of wild rice)”, it will be better that the authors show the phylogenetic tree including all related species (or most species) here especially when they want to talk about the Rf/rf origins and parallel evolution.

Line197: “from *O. rhizomatis* (CC-genome) to *O. nivara*”, please revise this statement.

Line162: “parallel evolution of Rf4”, I didn’t find clear evidence showing that the Rf4 experience parallel evolution after checking the Line163-Line205. Parallel evolution refers those two or more lineages evolved in similar ways, so that the evolved descendants are as similar to each other as their ancestors were. Moreover, whether a particular case is considered parallel may depend on how far back one looks in their ancestral lines. Since the author conclude that “thus the “a” locus likely represents the

ancestral sequence of Rf4 (Anc-Rf4a1)", the reader may feel that all alleles of Rf may originate from this Anc-Rf4a1.

Line267-269: "To investigate the relationship between haplotypes of Rf4 and rf4 and the application of major restorer lines in China, we obtained data about the planting areas of hybrid rice varieties". In general, the planting areas of a specific hybrid rice variety show heavily dependency on multiple factors/processes such as the genetic background, the vigor of the pollen itself, resilience in the abiotic environment, or even market activity/consumer behavior, while here we only discuss genetic factors, then the author should discuss is there any possibility that the other linked genes of Rf may play a role in contributing to the planting areas of this specific haplotypes.

Line322-323: "the replacement of rf4i derived from indica type maintainer lines is a precondition for the creation of modern CMS-WA lines", the author just mentioned that the footprints of how modern CMS-WA lines was cultivated with regarding to integrate rf4i, however, the author didn't show some clue to explain why rf4i is better than rf4a-like (rf3) with regarding to breed modern CMS-WA lines? Interestingly, the pseudogenized rf4i seems do better job than the one rf4a-like gene which show the power of artificial selection with regrading to the CMS. Thus, there is a possibility that, if people totally remove the rf4i gene in sterile line or maintain line, the CMS system would be more efficient. Actually, in a Brassica napus GMS system(7365ABC), the rf gene is a null gene which means no corresponding recessive rf gene at all. This may explain that why GMS is more stable than CMS in general.

Reviewer #4:

Remarks to the Author:

Key results

It has been shown that the fertility rate is significantly improved by the copy number of Rf4. However, even with two copies of Rf4, the fertility rate remains around 70%, indicating that there is still room for improvement in fertility rate.

Validity

I believe that the interpretation of the data and the validity of the conclusions are sufficient. However, in terms of the robustness, I judge that it is insufficient from the viewpoint of biological replicates, which will be mentioned later.

Significance

I believe that this paper is highly important for hybrid rice breeding. To further enhance its importance, it would be desirable to demonstrate the possibility that the fertility rate can be further improved with three copies of Rf.

Data and methodology

Regarding Fig.2b, it would be desirable to increase the number of samples to clearly demonstrate the relationship between the number of introduced individuals and fertility rate or starch accumulation ratio. It is also important to show that the function of the target gene was evaluated stably by using multiple individuals, as there are cases where individuals do not recover even after introducing a functional Rf.

For Fig.4, it would be desirable to add photos and data that demonstrate the factors other than pollen viability that may explain the low fertility rate, as there is a possibility that other factors may be affecting the fertility rate, such as the opening of the anthers, especially in the case of Jin23A×ZSRf4I.

Regarding Fig.5, if Rf4a/b M 723 -knockout plants consist of only a1 and a2, it may not be considered as biological replicates due to the small number of individuals. Therefore, it is desirable to increase the number of individuals.

Analytical approach

In terms of biological replicates, the paper may be considered weak. It would be desirable to increase the number of individuals or samples to improve the robustness of the results.

Suggested improvements

It is desirable to use three or more individuals for the evaluation of fertility rate in transformed plants to improve the robustness of the results (Fig.5).

Clarity and context

I understand that it can be very difficult to compare the results due to the large number of strains and their long names. It may be helpful to use a standardized naming system or abbreviations to make it easier to compare and understand the results.

References

3: It is mentioned that CMS-WA is the most widely used, but there is no mention of CMS-HL and CMS-BT. It would be desirable to gather additional information on these CMS if possible.

4: It is stated that CMS-WA has a 90% share, but there is no citation in the Introduction section where this is mentioned. If possible, please provide the source of this information.

5: It is stated that the share of CMS-WA is decreasing. If there is more recent information on the share rate, it would be desirable to update the citation accordingly.

6-8: It may be sufficient to cite only the reference in sentence 8.

9: There may be better reference sources available for citation.
<https://www.cabdirect.org/cabdirect/abstract/19826745461>

41: The reference may not be necessary as it shows that the CMS/Rf system is not widely used in wheat.

44: It may not be possible to determine the relevance of the reference as the current institution does not have access to it.

Your expertise

I am not an expert on CNV (copy number variation).

Point-to-point responses for reviewers' comments

Reviewer #1:

Hybrid rice make great contributions for ensuring the food security of the world, especially in China. CMS-WA type hybrid rice is first found and applied in China, and account for more than 80% three-line hybrid rice. Here, authors detailed analyzed the *Rf4* region in wild rice and cultivated rice, and raised the hypothesis that *Rf4* locus emerged earlier than the CMS-WA gene *WA352*. They cloned these eight haplotypes of the *Rf4* locus and verified that the two-copy *Rf4* haplotype has stronger fertility restoration ability compared with single-copy haplotypes. The conclusion is that dosage effect of *Rf4* is quite important for breeding the elite restorer lines. This research will increase our understanding of CMS/*Rf* systems and benefit crop breeding for utilization of heterosis in CMS-WA system. I strongly recommended for publication.

Response: Thank you for overall positive comments.

Minor questions:

1) Did the authors analyze the distribution of the wild rice, landraces? Maybe it is important to explain the origin of *WA352* and *Rf4*.

Response: Yes, we did analyze the distribution of wild rice, landraces in addition to cultivars; the data were shown in Fig. 4 and Supplementary Tables 2-4 in the original submission.

Given that Fig. 4 and Supplementary Fig. 3 of the previous version had similar key information, we have rearranged these figures to simplify the presentation for a better understanding Fig. 2 in the revised version.

In terms of the origins of *WA352* and *Rf4*, our previous study has shown that *WA352* is a newly formed gene derived from the common wild rice (*O. rufipogon*, AA-genome), which is mainly distributed in south/southeast Asia and northern Australia (Tang et al., 2017, *Cell res.*; Wing et al., 2018, *Nat. Rev. Genet.*). In current study, we showed that *Rf4/rf4* of rice cultivars emerged in EE- and CC-genome ancient wild rice species, which are found in northern Australia, and south/southeast Asia, respectively (Roulin et al., 2008, *Plant J.*), suggesting its evolutionary conservation. We mentioned these details in the Discussion section in the revised version (Page 15, lines 406–409).

2) In figure 5c, what is *UFCl*? Please clarified it in Materials and Methods in detail.

Response: *UFC1* (encoding a homolog of UFM1-conjugating enzyme 1) serves as an internal reference for nuclear genes in this study. As requested, we have supplied the detailed information on *UFC1* (encoding a homolog of UFM1-Conjugating Enzyme 1, GenBank number: AK059551), which is a constitutive expression gene with medium levels in the Methods (Page 18, lines 491–493) and the legend of Fig. 4 (Page 30, lines 748-750).
3) In supplemental figure 5e, the maximum of ordinate should be 100. I suggest it can be revised as 5a and 5c

Response: The reviewer is right. We apologize for this carelessness and have corrected the maximum of the ordinate to 100 in new Supplementary Fig. 7e in this revision.

Reviewer #2:

1) Key results

This is a report of rice *Rf4* that restores pollen fertility to CW-CMS plant. Variants of rice *Rf4* are considered as haplotypes consisting of one to several PPR protein-coding genes. The strength of *Rf4* haplotypes to restore pollen fertility is explained by gene dose effect of the specific copy of the PPR gene. Strong haplotype is favored in rice hybrid varieties.

2) Validity

I understand the high plasticity of *Rf4*. But the manuscript also claims purifying selection acting on *Rf4*. How are these be reconciled?

Response: Based on *Rf4* sequencing data for 720 accessions from 241 wild rice species and 168 landrace and 311 modern Asian and African cultivars, we identified numerous SNPs and structural variations in the sequences of *Rf4* locus, consistent with the high plasticity of the *Rf4* locus.

In response to the question of this reviewer, we consulted with Prof. Yalong Guo of the Institute of Botany, Chinese Academy of Sciences, an expert on plant evolution. He agreed with the reviewer that the concept of purified selection may not be applicable here only based on the *dN/dS* ratio of *Rf4* in this story. Given that our story mainly focuses on the relationship between CNV-mediated gene dosage and human selection of *Rf4* in three-line hybrid rice breeding, the data of *dN/dS* ratio, which is a parameter for assessing natural selection of *Rf4* locus, seems unnecessary. Taking the reviewer's Comment 6 into

consideration, we therefore have deleted the unnecessary data on the dN/dS ratio and do not claim the possible purifying selection on the *Rf4* locus in this revision.

3) Significance

This is the first report of gene dose effect of rice *Rf*. But not in crop. Therefore, line 92 is invalid. Gene dose effect of *Rf* has been reported in <https://doi.org/10.1007/s00122-018-3211-6>. The molecular basis of gene dose effect of *Rf* has been reported in <https://doi.org/10.1186/s12870-020-02721-9>. Relationship between *Rf* variant and plant breeding has been reported in <https://doi.org/10.1007/s11032-013-9854-8>.

Response: Gene duplication or copy number variation (CNV) are common in genomes. In this study, we found the *Rf4* expression level was associated with *Rf4* CNV and validated the biological significance of the CNV-mediated gene dosage in restoration of CMS-WA by experimental data. Our conclusion is that the more copy of *Rf4* results in the stronger capacity in restoration of CMS-WA.

Among three papers mentioned by the reviewer, the first paper entitled “Identification and characterization of a semi-dominant *restorer-of-fertility 1* allele in sugar beet (*Beta vulgaris*)” published in 2019 in *Theor. Appl. Genet.*, which demonstrated that the function of a semi-dominant *Rf1* in sugar beet is associated with gene dosages of homozygote and heterozygote: the homozygote produces full fertility, while the heterozygote restores partial fertility. However, in F₁ hybrids for field production, the related *Rf1* is present as a heterozygote.

The second paper entitled “The molecular basis for allelic differences suggests *Restorer-of-fertility 1* is a complex locus in sugar beet (*Beta vulgaris* L.)” published in 2020 in *BMC Plant Biol.*, which demonstrated that sugar beet *Rf1* locus has differentiated into dominant, semi-dominant and recessive alleles, which produced gene dosage effect on the amount decrease of the 250-kDa preSATP6 protein complex.

The third paper entitled “Identification of the predominant nonrestoring allele for Owen-type cytoplasmic male sterility in sugar beet (*Beta vulgaris* L.): development of molecular markers for the maintainer genotype” published in 2013 in *Mol. Breeding*, which analyzed organizational variation of the *Rf1* locus showing that *Rf1* is a multi-allelic locus, and developed a set of molecular markers to enrich maintainer genotypes in sugar beet germplasms.

Therefore, our finding of *Rf4* gene dosage is based on copy number variation, while that of sugar beet *Rf1* is based on heterozygote/homozygote and expression levels among differentiated alleles. Based on the reviewer's comments, we toned down our statement (Page 4, lines 93, 94), and cited one of the most relevant papers in this revision (Ref. 35, Page 22, lines 604–606).

4) Analytical approach

On the analysis of *dN/dS* ratio, care should be needed. Especially selecting samples.

Response: Thank you for reminding us of this important point. In consideration of the reviewer's Comments 2 and 6, we have deleted the unnecessary data and text related to *dN/dS*.

5) Suggested improvements

The manuscript is inflated by unnecessary data, and streamlining the manuscript could make it better.

Response: Thank you for this constructive suggestion. We have accepted the reviewer's suggestions and deleted unnecessary *dN/dS* data and related text to streamline our manuscript for ease of understanding.

6) Clarity and context

The manuscript is poor in presentation. It is difficult to follow the meaning. This is in part due to the ambiguous usage of terms. For example, genetic symbol is not consistent. I see *rf3* instead of *rf4* (line 225). Is *rf3* an allele of *rf4*? I think it is a kind of laboratory jargon. Please follow the rule of genetics. I also see confusion between 'allele' and 'haplotype' (e.g. line 176; this is just an example, and check throughout the manuscript). A number of symbols are used without definition. I cannot understand the content of lines 161 to 205 in part because of no instruction which 'allele' corresponds to the homologue in Fig. 3.

Response: We apologize for the complexity and inconsistency of *Rf4* terms, which is partly due to its high plasticity. After consulting a geneticist about the usage of terms, we have unified the terms to reduce the complexity. Given that *Rf4* is a complex locus consisting of one or two homologous *Rf4* open reading frame(s), we defined each *Rf4* ORF copy as a "variant" according to the copy position and sequence variation: the first copy of *Rf4* origination as "Copy-a" (for example *Rf4a^M*, *Rf4a^L*, *rf4aus*, *rf4i*, and *rf4j*), and the second *Rf4* duplication as "Copy-b" includes *Rf4b^M* and *rf4b*. One-copy or two-copy *Rf4*

variant(s) at the *Rf4* locus are defined as a “haplotype” (for example H1 = *Rf4a^M-Rf4b^M*, H2 = *Rf4a^I*, H6 = *Rf4b^M*, as shown in Fig.1b). To avoid confusion, we deleted unnecessary information of *Rf3*, another *Rf* gene on Chromosomes 1 for CMS-WA that is not involved in this work (Zhang et al., 1997, *Theor. Appl. Genet.*)

For ease of understanding, we simplified the arrangement in new Fig. 2 and new Supplementary Fig. 2, and rephrased the main text according to the reviewers’ comments (Pages 6, 7, lines 147–193).

7) References

Kato et al. (<https://doi.org/10.1038/sj.hdy.6801026>) should be cited for the structural variation of rice *Rf-1* on the chromosome 10 because this is the first to report the polymorphism of this locus.

Response: Thank you for the constructive advice. Rice Chromosome 10 harbors a *PPR* gene cluster, which includes at least 10 *PPR* genes. Both *Rf-1* and *Rf4* are in this cluster and the *Rf-1* locus consists of two functional *PPR*-type genes *Rf1a* and *Rf1b* for silencing of CMS-BT gene *orf79* (Wang et al., 2006, *Plant Cell*). Later, Kato et al. further reported the polymorphism of the *Rf-1* locus in *Oryza* species with AA genome (2007, *Heredity*). As requested, we have cited these references in this revision (Ref. 47, Page 23, lines 633, 634).

Reviewer #3:

The paper is a follow-up work of the classical CMS gene *WA352* controlling rice WA CMS. Based on former studies of this CMS system, the author tried to further explore the evolutionary trajectory/dynamics of the CMS restorer gene *Rf4*. The wet experiments are solid and show significant potential in improving the efficiency of this pollen control system. However, I am mainly concerned that the evidence of parallel evolution of *Rf4* shown in this study is not detailed and clear. The reliable evidence to show parallel evolution is that those species were under some specific biotic or abiotic selection pressure and gradually evolved similar phenotypes. For example, parallel evolution was typically reported in *Arabisopsis*, *Gasterosteus aculeatus*, and flying frog under which the selection pressure probably is climatic factors, freshwater/saltwater niches, and predation pressure,

respectively. However, the author did not discuss the potential driver of promoting the parallel evolution of *Rf4* as well.

The author composed some figures to show the correspondence relationship between the haplotypes and wild species/varieties. They avoided choosing the phylogenetic trees of those species to show the variance pattern of CNV and SV, making the evidence not explicit. Additionally, in Supplementary Table 3 and Supplementary Table 4, the author missed the information on the geographical distribution of those Landrace/Wild rice accessions and the main planting region of those cultivated rice.

Additionally, many narratives of language should be largely improved, such as “have important uses”, “robust fertility restorer (*Rf*) genes”, “first major *Rf*”, “by natural and artificial selection during domestication.”.

Response: We thank the reviewer for the very professional comments, and have tried our best to amend the manuscript accordingly.

Comments:

1) Line1: The “dosage” mentioned in the title seems ambiguous. The author may need to clarify whether it is transcriptional dosage or translational dosage based on their expression experiments, or just simply refer to two copies of it.

Response: We thank the reviewer for this rigorous comment. Our investigation showed that the CNV affects *Rf4* restoration ability. Increased copy number of *Rf4* increases the transcriptional dosage of *Rf4* (Figs. 4, 5) and likely the translational dosage too. To make it clear, we have replaced “dosage” with “copy number variation” in the title.

2) Line15-16: What do the authors really mean by “robust fertility restorer (*Rf*) genes”? Please make a clarification. “first major *Rf* for Wild-Abortive CMS,” what does “major” exactly mean? The gene frequency is highest or the ability to restore fertility is the strongest. Cite related paper at the end of line16.

Response: Thank you for the comment, which has helped us clarify the manuscript. “robust fertility restorer (*Rf*) genes” means a *Rf* gene with strong restoration ability for CMS lines. To make it clear, we have replaced “robust” with “strong” in the Abstract and main text.

“Major gene” means a gene individually has pronounced phenotypic effects, while “minor gene” only has a partial or weak genetic effect on a quantitative phenotype. The

“major” here refers to the *Rf* gene with strong ability in fertility restoration. There are two major *Rf* genes, *Rf3* and *Rf4*, accounting for CMS-WA restoration. Our group have isolated *Rf4* (Tang et al., 2014, *Mol. Plant*), while *Rf3* has not been cloned yet.

We have cited related papers in the part of “Introduction” (Ref. 15, Page 20, lines 557–559).

3) Line17-18: “However, the formation of the CMS-WA/*Rf* system and *Rf4* evolution during domestication remain elusive.” The word “formation” coordinates well with “during domestication”; however, “*Rf4* evolution” does not coordinate with “during domestication” since evolution emphasizes a natural process but not domestication by humans. The author should think about the difference between “domestication” and “artificial selection.” I prefer to use “artificial selection” here than “domestication” since humans would not expect to domesticate a male sterile phenotype.

Response: The reviewer is right. We have accepted the reviewer’s suggestion and replaced “domestication” with “human selection” in this revision.

4) Line29: Please remove “for increased yield” unless you have preliminary evidence in this paper.

Response: We have removed “for increased yield” in this revision.

5) Line36: The author should cite the pioneer paper of each type of CMS.

Response: We have cited the pioneer papers of each type of CMS (Refs. 4–6, Page 19, lines 533–539).

6) Line37-38: Could the author please clarify in which years range and which regions (or countries) CMS-WA occurs in approximately 90% of hybrid rice? “Therefore,” this seems illogical here. The author should revise it since CMS-WA occurs in approximately 90% of hybrid rice. It’s great enough compared to other systems. How the author deduces that “Therefore, understanding the evolution and domestication of related fertility restorer (*Rf*) genes provides important information for facilitating hybrid rice breeding.” Additionally, the author should check the citation 4 and 5 here for the most reliable resource of this information.

Response: We have revised the contents according to the reviewer’s suggestion and confirmed that citations 4 and 5 (Refs. 7, 8) provide reliable information (Page 19, lines 540–544).

7) Line 41-43: "has undergone ~15 million years of evolution" could be revised to "has undergone significant diversification over the course of evolution, with many of these species diverging even within the past 1 million years."

Response: We have revised this sentence according to the reviewer's suggestion (Page 2, lines 50, 51).

8) Line 44: The citation "6-8" could be replaced with a more relevant reference that supports the statement. For instance, the author could cite the pioneer paper that reveals this view by abundant taxonomy evidence or a paper with most molecular evidence mainly underlying this statement.

Response: We have replaced citations 6–8 with the pioneer papers that reveal this view by abundant taxonomy evidence (Glaszmann, J.-C. 1987, *Theor. Appl. Genet.*; Oka HI. 1988, *Tokyo: Jpn. Sci. Soc. Press*; Garris, A.J. et al., 2005, *Genetics*) (Refs. 11–13, Page 20, lines 550–554).

9) Line92: check this paper, "Evolution of the Brassicaceae-specific MS5-Like family and neofunctionalization of the novel MALE STERILITY 5 gene essential for male fertility in *Brassica napus*" (2020), they reported a restorer gene could inhibit another allele in *B. napus* in a dosage-dependent manner.

Response: We have carefully read the paper, in which the *Rf* gene *BnMS5^d* can inhibit another allele *BnMS5^a* in a dosage-dependent manner in *B. napus*. However, in our study the CNV-mediated *Rf4* gene dosage increases the ability of fertility restoration in rice. We have cited the recommended paper (Ref. 37, Page 22, lines 610–612).

10) Line111: The author could clarify how they confirmed that these genomes have the most reliable assembly within these *Rf* regions. The author could indicate the version and cite the papers and resource links.

Response: Based on these reference genomes, we designed primers, amplified the target fragments, and sequenced amplicons of the *Rf4* locus, which matched the respective reference genomes. As suggested, we have cited the latest published papers and resource links (Ref. 23, Page 21, lines 576, 577; Refs. 38–40, Page 22, lines 613–618).

11) Line118: The author could indicate whether there is any functional evidence showing that the *rf4i* gene was pseudogenized due to a premature termination codon.

Response: *rf4i* exists in the CMS-WA line (harboring CMS gene *WA352c*), suggesting its lack of male fertility restorer function. As suggested, through *ORF* prediction and sequencing, we found a new *PPR* gene encoding 815 amino acids downstream of the *rf4i* gene in CMS-WA and maintainer lines. Here we made a schematic diagram to show the *rf4i* gene is a pseudogene in fertility restoration due to a premature stop codon.

12) Line121: “are identical”, please indicate the % of identity and similarity between the two genes. “as are the regions 7.5 kb upstream of the start codon and 1.5 kb downstream of the stop codon”, it’s hard to believe that the 7.5 kb upstream and 1.5 kb downstream are identical. Usually, the duplicate at least experiences genetic drift and have some synonymous SNP.

Response: The reviewer is right. We did find SNPs in the 7.5-kb upstream region and the 1.5-kb downstream region of *Rf4a^M* and *Rf4b^M* in sequence alignment. The similarity of them is 98.4% and 99.7%, respectively (Page 5, line 130). However, the coding sequence similarity between *Rf4a^M* and *Rf4b^M* is indeed 100% in MH63 and SH498 cultivars. We have added percentage of similarity in Page 5, line 130.

13) Line124: “Therefore,” illogical problem.

Response: The “Therefore” has been replace to “Then” (Page 5, line 133)

14) Line126: Please indicate the basic information and the reason why chose these 304 cultivars. How could the 304 cultivars figure out all the alleles by the primers based on those SNPs which only come from the four publicly available rice genome information?

Response: These 304 Ascian cultivars were randomly chosen from different rice planting regions in China, representing a genetic core collection. The source or distribution for the 304 cultivars were added in Supplementary Table 2. The optimal primers designed for genotyping in this study were not only based on those SNPs from the four public rice genome information, but also on screening of many candidate primers by sequencing and alignment of amplicons at the *Rf4* locus.

15) Line127: since Line121-122 indicate that “as are the regions 7.5 kb upstream of the start codon and 1.5 kb downstream of the stop codon”, how can the author design the site-specific PCR primers based on SNP?

Response: The same specific forward primer F1 locating within the promoter region is common for PCR of Copy-a and Copy-b (Fig. 1a). The specific reverse primers a-R and b-R with SNPs between them are located outside of the 1.5-kb downstream region to distinguish Copy-a and Copy-b (Fig. 1a). Primers of F2 (within the promoter) and i-R (at the downstream of the stop codon of *rf4i*) are specific for the *rf4i* allele (Fig. 1a).

16) Line151-152: “these motifs are important for the fertility restoration of CMS-WA”, should be “amino acid substitutions at the PPR13, PPR14, and PPR15 motifs are important for the fertility restoration of CMS-WA”. What are the variances of other PPR motif (PPR1-PPR12.) between those *Rf4* and *rf4*?

Response: Thank you for the suggestion. We changed the wording to “amino acid substitutions at these PPRs motifs are important for the fertility restoration of CMS-WA” on Page 8, lines 207, 208. The variants of other PPR motifs between those *Rf* and *rf* are shown in Supplementary Fig. 3.

17) Line172: I strongly suggest that the author make an additional Figure showing the aa BLAST among all those putative homologs within Poaceae genomes.

Response: Thank you for your constructive advice. We have performed aa BLAST among all those putative homologs within *Poaceae* genomes. Since it is difficult to show all homolog information in one figure, we have presented the detailed information in an Excel table as a compromise of figure (Supplementary Data 3).

18) Line174: “identity”, please indicate the % here.

Response: To make it consistent, we have unified the usage of “similarity” in the manuscript and added the percentage accordingly in the text (Page 6, line 160).

19) Line692: It’s hard to sort the detail by looking at this circle tree.

Response: The phylogenetic tree of Fig. 3 has been changed to a new style in new Supplementary Fig. 2.

20) Line705: I am concerned with the Fig. 4 either because it’s unclear to show the evolutionary events such as “expansion” and “Hybridization, gene gain and loss” or because it is non-compliant on exhibiting the evolutionary time scale (or no clear

evolutionary time of all wild/cultivated rice). For example, the author exhibits three panels in the figure 4 to show the 3 stages: ancestral, wild rice and cultivated rice. I am not sure whether the “expansion” between ancestral and wild rice means more copy in each wild rice species or more different alleles originated. If it’s the former, we didn’t find there is more *Rf4/rf4* copy in each wild rice such as *rf4i/rf4j/rf4aus* than in ancestral, if it’s the latter, then this process is not expansion. Additionally, in “wild rice” panel, which species divergent time do the allele *rf4a/rf4b* or the allele *Rf4a/Rf4b* originated? I strongly suggest the author to implement an alternative by incorporate the phylogenetic tree into the Figure 4 (left panel is phylogenetic tree, right panel is the structure of alleles). BTW, for the Supplementary Fig. 3, the author didn’t choose the two species: *O. barthii* and *O. glaberrima*, which originated between *O. glumaepatula* and the ancestor of *O. nivara* and *O. rufipogon* (Stein, Nature genetics 2018). I would expect that there will probably be more interesting CNV and SV within this branch and it would be helpful for the author trace *Rf4*. The author could just simply check the genome sequence of *Rf4* region.

Response: Thank you for the constructive suggestion. Based on reviewers’ comments, information on the distribution of *Rf4* haplotypes in major *Oryza* species has been integrated into Fig. 4 to achieve a new Fig. 2 for ease of understanding in this revision.

The “expansion” in the *Rf4* locus means emergence of novel variants, duplication, and recombination to generate different haplotypes in the *Oryza* species. Therefore, the increase of *Rf4* copy and allelic variants may occur in parallel and randomly in the *Oryza* genus.

According to our study, the ancestral *Rf4* variant (*Anc-Rf4*) first appeared in the “Copy-a” location, but not “Copy-b” site in the GG-genome wild rice (*O. meyeriana*), which is regarded as the oldest species in *Oryza* genus. Thus, the “Copy-a” variant likely represents the ancestral sequence of *Rf4*. During evolution, *Rf4* locus might have undergone sequence variation, recombination and rearrangement, leading to gain and loss (such as H6 haplotype) of *Rf4* copy number at Copy-a and Copy-b to expand the *Rf4* haplotypes.

According to the reviewer’s suggestion, we have optimized the evolutionary trajectory of *Rf4* variants and haplotypes in original Fig. 4, and included a panel showing distribution information of haplotypes in major *Oryza* species in a new Fig. 2, making it easy to understand and reader-friendly.

As suggested, we have checked the genome sequence of *Rf4* locus in *O. longistaminata*, *O. barthii*, and *O. glaberrima*. However, no a *Rf4* variant sequence was detected in Copy-a or Copy-b, indicating that these three species may possess considerable structural and sequence variation, leading to gene loss in the *Rf4* locus, or before the divergence of two independent lineages, the *Rf4* gene may pass through a genetic bottleneck event only in *O. rufipogon* and *O. nivara*, but not in *O. longistaminata* and *O. barthii*, and finally be fixed in Asian cultivars.

21) Line188: “in the *O. meyeriana* species (GG-genome, ancestor of wild rice)”, it will be better that the authors show the phylogenetic tree including all related species (or most species) here especially when they want to talk about the *Rf/rf* origins and parallel evolution.

Response: Thank you for the suggestion. We have presented related information in a new version of Fig. 2 in this revision, so the readers may easily understand the origin, evolution, and artificial selection of *Rf4/rf4* in *Oryza* species.

22) Line197: “from *O. rhizomatis* (CC-genome) to *O. nivara*”, please revise this statement.

Response: We have revised this statement.

23) Line162: “parallel evolution of *Rf4*”, I didn’t find clear evidence showing that the *Rf4* experience parallel evolution after checking the Line163-Line205. Parallel evolution refers those two or more lineages evolved in similar ways, so that the evolved descendants are as similar to each other as their ancestors were. Moreover, whether a particular case is considered parallel may depend on how far back one looks in their ancestral lines. Since the author conclude that “thus the “a” locus likely represents the ancestral sequence of *Rf4* (*Anc-Rf4a^l*)”, the reader may feel that all alleles of *Rf* may originate from this *Anc-Rf4a^l*.

Response: We have deleted the description of “parallel evolution” to tone down our claim in the revised manuscript.

24) Line267-269: “To investigate the relationship between haplotypes of *Rf4* and *rf4* and the application of major restorer lines in China, we obtained data about the planting areas of hybrid rice varieties”. In general, the planting areas of a specific hybrid rice variety show heavily dependency on multiple factors/processes such as the genetic background, the vigor of the pollen itself, resilience in the abiotic environment, or even market activity/consumer behavior, while here we only discuss genetic factors, then the author

should discuss is there any possibility that the other linked genes of *Rf* may play a role in contributing to the planting areas of this specific haplotypes.

Response: The reviewer is absolutely right. The planting areas of a specific rice variety do depend on multiple factors, but in terms of hybrid rice varieties, the fertility restoration ability of CMS is a restrictive factor for its application.

The *Rf4* locus is located in a hotspot of *Rf* genes on Chromosome 10. Except *Rf4*, at least other four *Rf* genes have been identified including that *Rf1a/Rf5* for CMS-BT/CMS-HL, *Rf1b* for CMS-BT (Wang et al., 2006, *Plant Cell*; Hu et al., 2012, *Plant Cell*), and *Rf19* for CMS-FA (Jiang et al., 2022, *PNAS*). These *Rf* genes may also contribute to the planting area of a specific hybrid rice variety. We have discussed this point in the Discussion section (Pages 14, 15; lines 399–404).

25) Line322-323: “the replacement of *rf4i* derived from indica type maintainer lines is a precondition for the creation of modern CMS-WA lines”, the author just mentioned that the footprints of how modern CMS-WA lines was cultivated with regarding to integrate *rf4i*, however, the author didn’t show some clue to explain why *rf4i* is better than *rf4a*-like (*rf3*) with regarding to breed modern CMS-WA lines? Interestingly, the pseudogenized *rf4i* seems do better job than the one *rf4a*-like gene which show the power of artificial selection with regrading to the CMS. Thus, there is a possibility that, if people totally remove the *rf4i* gene in sterile line or maintainer line, the CMS system would be more efficient. Actually, in a *Brassica napus* GMS system (7365ABC), the *rf* gene is a null gene which means no corresponding recessive *rf* gene at all. This may explain that why GMS is more stable than CMS in general.

Response: Thank you for the comments. As we mention in Comment 2 and response to Reviewer #2 in Comment 2, there are two major *Rf* genes, *Rf3* (locating at the short arm of Chromosome 1) and *Rf4*, for CMS-WA, thus the *rf4a*-like variant is not *rf3*. However, of the three recessive *rf4* alleles found in Asian cultivated rice, *rf4i* is present mainly in *indica* cultivars, *rf4j* mainly in *japonica* cultivars and *rf4aus* in circum-*aus* rice lines. No null type of *rf4* allele (absence of the *rf4* sequence at all) was detected in all the analyzed Asian rice cultivars. Since CMS-WA and maintainer lines were breed from certain *indica* cultivars, it is reasonable that *rf4i* is dominantly used in the three-line hybrid rice varieties (*indica* type).

Reviewer #4:

1) Key results

It has been shown that the fertility rate is significantly improved by the copy number of *Rf4*. However, even with two copies of *Rf4*, the fertility rate remains around 70%, indicating that there is still room for improvement in fertility rate.

Response: We thank the reviewer for this insightful comment. As we mentioned, there are two major *Rf* genes, *Rf3* and *Rf4*, for CMS-WA restoration. The spikelet fertility rate around 70% in the lines of this study all carrying a recessive *rf3* allele. Thus, in breeding of hybrid rice, the restorer lines must contain both dominant *Rf3* and *Rf4* alleles for better restoration, while those carrying the two-copy *Rf4* and dominant *Rf3* may possess stronger and more stable fertility restoration ability in the CMS-WA system.

2) Validity

I believe that the interpretation of the data and the validity of the conclusions are sufficient. However, in terms of the robustness, I judge that it is insufficient from the viewpoint of biological replicates, which will be mentioned later.

Response: We have repeated the experiments with other two biological replicates to support the robustness of the data (Figs. 4, 5 and Supplementary Figs. 5, 6). The conclusion remains solid.

3) Significance

I believe that this paper is highly important for hybrid rice breeding. To further enhance its importance, it would be desirable to demonstrate the possibility that the fertility rate can be further improved with three copies of *Rf4*.

Response: Thank you for your recognition on the importance of our work. Fertility is a fragile agronomic trait that is sensitive to abiotic and biotic stresses such as heat, cold, drought and disease. Therefore, even for normal rice cultivars, the pollen fertility usually does not reach 100%. Based on our data in Figs. 4, 5 and Supplementary Figs. 5, 6, the fertility rate may improve a little bit with one more copy of *Rf4*, but may not reach 100%. The restorer lines carrying two-copy *Rf4* and *Rf3* may have relatively high and stable restoration ability in hybrid rice production.

4) Data and methodology

Regarding Fig. 2b, it would be desirable to increase the number of samples to clearly demonstrate the relationship between the number of introduced individuals and fertility rate or starch accumulation ratio. It is also important to show that the function of the target gene was evaluated stably by using multiple individuals, as there are cases where individuals do not recover even after introducing a functional *Rf4*.

Response: Thank you for the suggestion. We have added other two individuals of five different transgenic lines to address your concern in Supplementary Fig. 4.

For Fig. 4, it would be desirable to add photos and data that demonstrate the factors other than pollen viability that may explain the low fertility rate, as there is a possibility that other factors may be affecting the fertility rate, such as the opening of the anthers, especially in the case of Jin23A×ZSRf4I.

Response: Thank you for the suggestion. We have added anther phenotypes of different lines in Figs. 4 and 5. The partially sterile anthers were thin and whitish in Jin23A×ZSRf4I (Fig. 4) and Jin23A/*Rf4*⁻ (Fig. 5) lines (carrying the one-copy *Rf4*), while the fertile anthers of Jin23A×ZSRf4M (Fig. 4), Jin23A/*Rf4*⁺*Rf4*⁺, Jin23A/*Rf4*⁺*Rf4*⁺×ZSRf4I (Fig. 5), which harbor two-copy *Rf4*, are pollen-filled and yellowish.

Regarding Fig. 5, if *Rf4a/b*^M-knockout plants consisted of only a1 and a2, it may not be considered as biological replicates due to the small number of individuals. Therefore, it is desirable to increase the number of individuals.

Response: Thank you for the suggestion. We have added other two different *Rf4a/b*^M-knockout mutants (*rf4a^m-rf4b^m*) in Supplementary Fig. 5b to make our conclusion solid.

5) Analytical approach

In terms of biological replicates, the paper may be considered weak. It would be desirable to increase the number of individuals or samples to improve the robustness of the results.

Response: Thank you for the suggestion. We have increased the number of individuals in different lines to confirm the copy dosage effect of *Rf4* (Supplementary Figs. 5, 6) and improve the solidness of the data.

6) Suggested improvements

It is desirable to use three or more individuals for the evaluation of fertility rate in transformed plants to improve the robustness of the results (Fig. 5).

Response: Thank you for the suggestion. We have added other two individuals to improve the solidness of the data in Supplementary Figs. 5, 6.

7) Clarity and context

I understand that it can be very difficult to compare the results due to the large number of strains and their long names. It may be helpful to use a standardized naming system or abbreviations to make it easier to compare and understand the results.

Response: In response to this suggestion, we have unified the naming and increased the number of individuals or samples to improve the quality of our manuscript.

8) References

Ref3: It is mentioned that CMS-WA is the most widely used, but there is no mention of CMS-HL and CMS-BT. It would be desirable to gather additional information on these CMS systems if possible.

Response: Thank you for the suggestion. We have added some information on CMS-HL and CMS-BT in the text (Page 2, lines 43, 44).

Ref4: It is stated that CMS-WA has a 90% share, but there is no citation in the Introduction section where this is mentioned. If possible, please provide the source of this information.

Response: Sorry for this error. We have revised this statement and provided the source of this information (Page 2, lines 42–44).

Ref5: It is stated that the share of CMS-WA is decreasing. If there is more recent information on the share rate, it would be desirable to update the citation accordingly.

Response: The reviewer is right. The planting area of CMS-WA is decreasing, due to its complexity and inflexibility of three-line system compared with newly developed two-line system based on thermo-sensitive genic male sterility. Compared to CMS-HL and CMS-BT, CMS-WA is predominant in three-line hybrid rice production in China (1983-2012) (Refs. 7, 8, Page 19, lines 540–544). Based on the data from The China Rice Data Center (<http://www.ricedata.cn/>), the planting area of CMS-WA remained stable since 2012.

Ref6-8: It may be sufficient to cite only the reference in sentence 8.

Response: We have provided the source of this information (Refs. 11–13, Page 20, lines 550–554).

Ref9: There may be better reference sources available for citation.

Response: Thank you for the suggestion. We have cited the source paper (Ref. 4, Page 19, lines 533, 534).

Ref41: The reference may not be necessary as it shows that the CMS/*Rf* system is not widely used in wheat.

Response: We have deleted this reference.

Ref44: It may not be possible to determine the relevance of the reference as the current institution does not have access to it.

Response: We have replaced this reference with Ref. 28 in this revision (Page 21, lines 589, 590).

Reviewers' Comments:

Reviewer #1:

Remarks to the Author:

Thanks for addressing my concerns. I recommend it for publication in this revision.

Reviewer #2:

Remarks to the Author:

Lines 129 and 131: 'Identity' may be better than similarity.

Line 371: Asian. Please check.

Line 428: Five? I see the authors name four markers here.

Line 535: Author names are missing.

Reviewer #3:

Remarks to the Author:

The author effectively addressed and resolved my comments regarding this article.

Reviewer #4:

Remarks to the Author:

All of the questions I raised were answered appropriately. As a further development of this paper, I became interested in how the expression pattern of WA352 differs depending on the copy number and how it affects fertility rates. I hope for further research and development by the writing group.

Point-to-point responses for reviewers' comments

Reviewer #1 (Remarks to the Author):

Thanks for addressing my concerns. I recommend it for publication in this revision.

Response: Thank you so much for recognizing our study.

Reviewer #2 (Remarks to the Author):

Lines 129 and 131: 'Identity' may be better than similarity.

Response: Thank you for the comment. We have replaced 'Similarity' with 'Identity'.

Line 371: Asian. Please check.

Response: Sorry for this error, we have revised to 'Asian'.

Line 428: Five? I see the authors name four markers here.

Response: Here did have five markers (*Rf4*, Copy-a, Copy-b, *rf4a* and *rf4a/b*).

Line 535: Author names are missing.

Response: Thank you for reminding this point. In last revision, we cited the original paper of this current reference, however, this reference only included the name of institution but lacked author names (1977, *Acta Genetica Sin.* 4, 219–227) (See below), so this reference has not Author names.

BREEDING OF MALE-STERILE LINES OF RICE WITH THEIR MAINTAINERS AND RESTORERS OF FERTILITY BY MEANS OF HYBRIDIZATION OF WILD RICES FROM SOUTH CHINA WITH CULTIVARS

Research Laboratory of Genetics, Wuhan University, Division of Eice Culture, Hupeh Provincial Institute of Agricultural Research

By means of crossing two forms of wild rice (*Oryza sativa* L. f. spontanea) : "red-awned" and "Tengqiao local" from the Island of Hainan as the female parent with cultivars as the male and repeated back-crossing we have obtained several male-sterile lines of rice. We also succeeded in selecting some sterility-maintainers and fertility-restorers for them respectively to facilitate the utilization of heterosis in rice. It is not worthy that the two forms of wild rice have different kinds of cytoplasm and they respond differentially to test crosses. Therefore the male-sterile lines derived from the two wild rices have quite different restorer lines. From genetical and cytological studies it is evident that male sterile

Reviewer #3 (Remarks to the Author):

The author effectively addressed and resolved my comments regarding this article.

Response: Thank you so much for constructive suggestions for improving the quality of our manuscript.

Reviewer #4 (Remarks to the Author):

All of the questions I raised were answered appropriately. As a further development of this paper, I became interested in how the expression pattern of *WA352* differs depending on the copy number and how it affects fertility rates. I hope for further research and development by the writing group.

Response: Thank you for your recognition and encouragement. As the reviewer requested, we will explore the interesting questions in future and hope to publish related story soon.